# CARIBIC-AMS: A fully automated aerosol mass spectrometer for operation on routine passenger flights (IAGOS-CARIBIC): Instrument description and first flight application in the UTLS

Johannes Schneider[1], Christiane Schulz[1,2,3], Florian Rubach[1,2,4], Anna Ludwig[1], Jonas Wilsch[1], Philipp Joppe[1,5], Christian Gurk[6], Sergej Molleker[6], Laurent Poulain[2], Florian Obersteiner[7], Torsten Gehrlein[7], Harald Bönisch[7], Andreas Zahn[7], Peter Hoor[5], Nicolas Emig[5], Heiko Bozem[5], Stephan Borrmann[1,5], Markus Hermann[2]

[1]Particle Chemistry Department, Max Planck Institute for Chemistry, Mainz, 55128, Germany
[2]Leibniz Institute for Tropospheric Research, Leipzig, 04318, Germany
[3]Now at: Leibniz-Zentrum für Agrarlandschaftsforschung, 15374 Müncheberg, Germany
[4]Now at: Climate Geochemistry Department, Max Planck Institute for Chemistry, Mainz, 55128, Germany
[5]Institute for Physics of the Atmosphere, Johannes Gutenberg University, Mainz, 55128, Germany
[6]Instrument Development Group, Max Planck Institute for Chemistry, Mainz, 55128, Germany
[7]Institute of Meteorology and Climate Research, Karlsruhe Institute of Technology (KIT), Eggenstein-Leopoldshafen, 76344, Germany

*Correspondence to*: Johannes Schneider (johannes.schneider@mpic.de)

**Abstract.** In this study, we present a fully automated aerosol mass spectrometer (AMS) that is operated during regular passenger aircraft flights in the CARIBIC (Civil Aircraft for the Regular Investigation of the Atmosphere Based on an Instrument Container) module of the European Research Infrastructure IAGOS (In-service Aircraft for a Global Observing System - www.iagos.org). The instrument, termed CARIBIC-AMS, is able to measure the mass concentration of non-refractory aerosol species, namely sulfate, nitrate, ammonium, and organics, in a particle diameter range of approximately 50 – 800 nm. The main challenge was the mechanical and electrical redesign of a commercial instrument in order to comply with the operating and safety requirements in the IAGOS-CARIBIC container-laboratory before and during flight. In the container-laboratory, the instrument has to operate fully autonomous, typically during four consecutive long-haul flights à 10 hours. The CARIBIC-AMS weighs 74 kg, has a volume of 0.16 $m^3$ and consumes 360 W of electrical power during regular operation. Due to the short time for evacuation of the vacuum chamber to sufficiently low pressures before measurement, detection limits are higher during regular flights than during ground operation and were determined to be 0.035 µg $m^{-3}$ STP (sulfate), 0.055 µg $m^{-3}$ STP (nitrate), 0.69 µg $m^{-3}$ STP (organics), 0.38 µg $m^{-3}$ STP (ammonium), and 0.022 µg $m^{-3}$ STP (chloride) for a time resolution of 30 seconds. These values represent typical averages under flight conditions and refer to a collection efficiency of 0.5. Since the IAGOS-CARIBIC project aims for climatological, regular, long-term data, longer data averaging times are possible, thereby lowering the detection limits by the square root of the number of averaged data points. Data validation, calibration and instrument characterization were conducted by means of laboratory-based comparisons with existing, established aerosol mass spectrometers. Here we report on the details of the automation, the instrument characterization as

well as first in-flight data measured in the upper troposphere and lower stratosphere during two IAGOS-CARIBIC flights and during the TPEx (Tropopause composition gradients and small-scale mixing Experiment) field campaign, conducted in 2024 using a Learjet as research aircraft over Northern Europe.

## 1 Introduction

Aerosol particles play an important role in atmospheric physics and chemistry. They contribute to direct and indirect radiative climate forcing, influence (as cloud condensation nuclei or ice nucleating particles) the hydrological cycle, supply nutrients to oceanic and terrestrial ecosystems, influence heterogeneous atmospheric chemistry, lead to adverse health effects, and reduce visibility (e.g., Andreae and Rosenfeld, 2008; Rosenfeld et al., 2008; Heintzenberg and Charlson, 2009; Boucher et al., 2013; Guerreiro et al., 2018; Bellouin et al., 2020; Li et al., 2022).

Although the abundance of particulate mass is much lower in the upper troposphere and lower stratosphere (UTLS) compared to the atmospheric boundary layer, the UTLS aerosol still significantly influences the radiative balance of the Earth. This effect is caused by the longer lifetime of particles in the UTLS compared to lower altitudes. Moreover, aerosol particles in this region provide surface area for heterogeneous chemistry and influence the formation of ice clouds (Krämer et al., 2009) which impact the Earth´s radiation budget (Liou, 1986; Yang et al., 2015; Yi et al., 2016; Järvinen et al., 2018). The UTLS region represents also a source region of the stratospheric aerosol (Brock et al., 1995; Borrmann et al., 2010; Weigel et al., 2011; Williamson et al., 2019), which is not only of relevance for stratospheric ozone destruction (Hofmann and Solomon, 1989). Solomon et al. (2011) showed that the stratospheric aerosol background in times of volcanic quiescence is more variable than previously thought. There is an increasing trend in the stratospheric aerosol leading to a significant radiative forcing, mostly neglected by global climate models. The discovery of the Asian Tropopause Aerosol Layer (ATAL) by Vernier et al. (2011), its subsequent characterization by means of in-situ measurements (Höpfner et al., 2019; Appel et al, 2022), and the much stronger than previously thought radiative impact of volcanic aerosols in the UTLS (Andersson et al., 2015; Ridley et al., 2014) are two additional examples for the importance, but also for the limited knowledge of the UTLS aerosol. In the past, it was widely assumed that UTLS aerosol particles consist mainly of binary sulfuric acid/water solutions, the former being formed from $SO_2$ oxidation (Rosen, 1971; Brock et al., 1995). Besides that, significant amounts of refractory material, most likely originating from meteorites, were found in the LS (Murphy et al., 1998, 2014; Curtius et al., 2005; Weigel et al., 2014; Schneider et al., 2021). Similarly, black carbon (soot) containing particles, originating from aircraft emissions or biomass burning, were measured with high frequency in the UTLS (Blake and Kato, 1995; Schwarz et al., 2010; Ditas et al., 2018; Solomon et al., 2022). In recent years, organic carbon (OC) particles have been frequently detected in the UTLS, e.g., by aerosol mass spectrometry (Murphy et al., 1998, 2014; Froyd et al., 2009; Schmale et al., 2010) and they seem to be the dominant aerosol fraction at 9-14 km altitude in tropical regions over the continents (Nguyen et al., 2008; Froyd et al., 2009; Andreae et al., 2018; Schulz et al., 2018; Yu et al., 2015). In the understanding of OC particles, there has long been a significant gap between observations, model results, and laboratory measurements (Heald et al., 2005, 2011; Chen et al., 2015; Tsigaridis et al., 2014),

but with significant improvements in recent years (Hodzic et al., 2020; Pai et al., 2020). Currently, global chemistry models have the tendency to overestimate primary organic aerosol and underestimate secondary organic aerosol (Hodzic et al., 2020). If the organic aerosol is transported as particles from the boundary layer, it is to be expected that such particles are highly oxidized and will have a high hygroscopicity (Zhang et al., 2007; Jimenez et al., 2009), but also an amorphous, glassy state

has been proposed for biogenic organic aerosol particles which would significantly influence their hygroscopic properties (Virtanen et al., 2010; Koop et al., 2011; Shiraiwa et al., 2017, 2011). Furthermore, recent observations (Schulz et al., 2018; Andreae et al., 2018; Mahnke et al., 2021; Weigel et al., 2021; Wang et al., 2022, Curtius at al., 2024) confirmed the suggestion by Kulmala et al. (2006) that volatile organic compounds can contribute to new particle formation and growth in the UT. Such newly formed particulate compounds are less oxidized, such that the formation and transport processes can be distinguished

by the degree of oxidation (Schulz et al., 2018). The degree of oxygenation of organic aerosol particles in the free troposphere was found to be underestimated by global chemistry models (Hodzic et al., 2020).

Especially for the UTLS, there is still a lack of representative aerosol measurements, which is mainly related to technical and financial issues. Research aircraft, capable of performing detailed investigations of high scientific value for aerosol process studies, are cost-intensive and thus respective campaigns cover only short time periods and small fractions of the globe.

Nevertheless, available data from numerous aircraft-based research campaigns have been used in recent years to better constrain global chemical transport models (e.g., Lou et al., 2020; Pai et al., 2020; Reifenberg et al., 2022; Norman et al., 2024). However, most of these data sets do not represent the full variability of atmospheric composition in the tropopause region. To obtain a more representative dataset, the Atmospheric Tomography Mission (ATom) used the NASA DC-8 research aircraft to four series of flights over the middle of the Pacific and Atlantic oceans. During these flights the DC-8 repeatedly

ascended and descended between the boundary layer and about 12 km in altitude to sample the whole troposphere (Brock et al., 2019, 2021; Williamson et al., 2019; Hodzic et al., 2020; Thompson et al., 2021; Froyd et al., 2022).

A complementary approach is followed in IAGOS-CARIBIC (In-service Aircraft for a Global Observing System - Civil Aircraft for Regular Investigation of the Atmosphere Based on an Instrument Container, www.iagos.org) which uses regular commercial passenger aircraft flights to conduct in-situ measurements of trace gases and aerosol particles (Brenninkmeijer et

al., 1999, 2007; Petzold et al., 2015). IAGOS-CARIBIC operates a modified cargo container with a set of fully automated instruments during two to four inter-continental measurement flights per month, thereby obtaining a large, regular data set on UTLS trace gases and aerosol particles.

Here, we describe the new CARIBIC-AMS (CARIBIC-Aerosol Mass Spectrometer) which was designed for operation in the IAGOS-CARIBIC container-laboratory to measure mass concentrations of non-refractory aerosol compounds of

submicrometer particles. We present the experimental setup, detail the necessary steps to achieve full automation of the instrument, document calibration procedures and quality checks, and show exemplary flight data, both from IAGOS-CARIBIC and from a separate aircraft-based field campaign.

## 2 Experimental

### 2.1 The IAGOS-CARIBIC container-laboratory

CARIBIC is in operation since 1997 on commercial passenger aircraft, first on a Boeing B767-300 of LTU International Airways, followed by a Lufthansa (LH) Airbus A340-600 (Brenninkmeijer et al., 1999, 2007) until March 2020. After the phase-out of the Lufthansa A340-600, the measurements will continue using a Lufthansa A350. Since November 1997 until March 2020 (with two larger breaks), two to four consecutive measurement flights per month have been carried out to 38 destinations in North, Middle and South America, Southern Africa, Europe, as well as South, South-East, and East Asia. In

2008, the CARIBIC project was merged with the MOZAIC project (Measurements of ozone, water vapor, carbon monoxide and nitrogen oxides by in-service Airbus aircraft) into the European Research Infrastructure IAGOS (Petzold et al., (2015). For each flight sequence, the IAGOS-CARIBIC container is prepared at the Karlsruhe Institute of Technology (KIT), Germany, transported to Munich airport, and mounted into the forward cargo compartment of the IAGOS-CARIBIC aircraft. Normally, the container remains on board taking data for a sequence of usually four consecutive flights. After the flight

sequence, the container is removed, transported back to KIT, where data are downloaded, instruments are removed from the container for calibration and maintenance, and samples are shipped to the respective institutes for laboratory analysis.

In March 2020, operation of the IAGOS-CARIBIC container was prematurely terminated, as Lufthansa phased out the CARIBIC Airbus A340-600 due to the severe air traffic reduction during the COVID-19 pandemic. Thus, the transition of the IAGOS-CARIBIC container-laboratory to the successor aircraft, an Airbus A350, had to start earlier than planned and the time

period without IAGOS-CARIBIC operation is longer than originally expected. The resumption of regular scientific flights is currently scheduled for the end of 2025.

In the IAGOS-CARIBIC container-laboratory configuration used on the A340-600, several aerosol parameters were measured: particle number concentration and particle size distribution (Hermann and Wiedensohler, 2001; Bundke et al., 2015; Hermann et al., 2016), black carbon mass concentration and coating (Ditas et al., 2018), elemental composition obtained by filter samples

for offline analysis using particle-induced X-ray emission (PIXE) and particle elastic scattering analysis (PESA) (Nguyen et al., 2006; Martinsson et al., 2014), and fluorescent particle concentration using a wideband integrating bioaerosol sensor (WIBS, Yue et al., in prep.). Trace gas measurements include ozone, water vapour, carbon monoxide, carbon dioxide, methane, reactive nitrogen, volatile organic compounds, and hydrocarbons (Brenninkmeijer et al., 2007; Schuck et al., 2009; Scharffe et al., 2012; Zahn et al., 2012; Baker et al., 2016; Stratmann et al., 2016). A dedicated trace gas and aerosol inlet is permanently

installed below the fuselage of the aircraft (Brenninkmeijer et al, 2007)

For future operation on the A350 aircraft, a new container-laboratory has been designed (https://www.caribic-atmospheric.com), manufacturing and integration of existing and new instruments is currently ongoing. Additionally, a new improved inlet system for gases and aerosol particles has been designed, manufactured and successfully deployed during a test flight with the lower A350 aircraft.


## 2.2 CARIBIC-AMS instrument description

### 2.2.1 General description

The CARIBIC-AMS is based on the commercially available "mini" aerosol mass spectrometer (mAMS), developed by Aerodyne Research Inc., which has already been deployed successfully in ground-based (Goetz et al., 2018) and aircraft-based
(Dingle et al., 2016; Vu et al., 2016) measurements. Similar to the time-of-flight aerosol chemical speciation monitor (ToF-ACSM, Fröhlich et al., 2013), the mAMS uses a compact time-of-flight mass spectrometer (C-ToF-MS). The difference between a ToF-ACSM and a mAMS is the chopper module that allows for sizing of the particles in the mAMS. The basic concept of an AMS has been described widely in the literature (Jayne et al., 2000; Drewnick et al., 2005; DeCarlo et al., 2006; Canagaratna et al., 2007) and will therefore not be reviewed in detail here, only the basic working principle is briefly described.
The particles are sampled from the ambient air through a critical orifice into an aerodynamic lens (ADL) which focuses the particles onto a narrow beam. The size range of particles transmitted and focused by this combination is about 50 – 800 nm in diameter (Liu et al., 2007). The particle beam is directed onto a standard vaporizer (e. g., Hu et al., 2017) made out of tungsten, operated at 600°C. Here the particles are flash vaporized, and the gas molecules are ionized by electron ionization. The ions are accelerated towards the time-of-flight mass spectrometer where they are separated by their mass-to-charge ratio.
In order to operate the mAMS under the above described IAGOS-CARIBIC conditions and to meet the safety requirements for aircraft operation, the mAMS had to be modified in several ways, as detailed below. The modified instrument is hereafter termed CARIBIC-AMS. A schematic of the instrument is shown in Fig. 1.

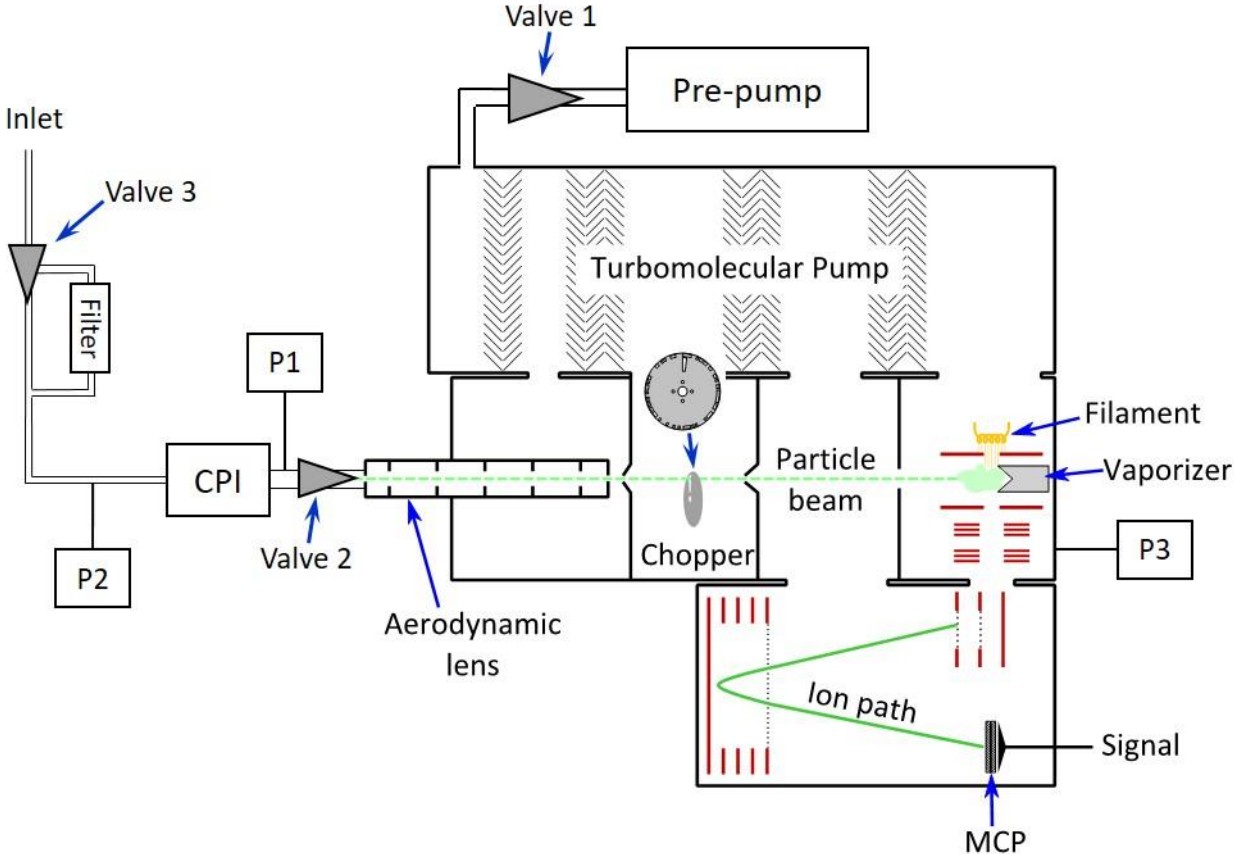

**Figure 1: Principal schematic of the CARIBIC-AMS, modified after Fröhlich et al. (2013). P1, P2, and P3 show locations of pressure sensors; CPI: Constant pressure inlet; MCP: Multichannel plate.**

### 2.2.2 Mechanical and electrical modifications

The original dimensions of the CARIBIC-AMS were the same as the ToF-ACSM and are 65 cm in length, 51 cm in width and 60 cm in height (Fröhlich et al., 2013). In order to be incorporated into the CARIBIC container, the original instrument and

additional equipment allowing for automatic operation had to fit into the existing rack structure of the CARIBIC container. Therefore, the CARIBIC-AMS was rebuilt and is now 60 cm long, 45 cm wide, and 63.5 cm high as highlighted in yellow in Fig. 2. Therefore, several electronic housings were combined into one new housing to save space.

To maintain constant pressure conditions in the aerodynamic lens, a constant pressure inlet (CPI, Molleker et al., 2020) was added in front of the lens. This type of CPI has already been used by our group in several aircraft deployment up to altitudes

of about 20 km (Schulz et al., 2018; Köllner et al., 2017; Höpfner et al., 2019; Schneider et al., 2021; Appel et al., 2022; Hünig et al., 2022). To allow for computer-controlled operation of the three plug valves (see Fig. 1), each plug valve is equipped with a servo motor. Further modification included replacement of flammable plastics by non-flammable material, partly 3-d printed,

addition of thermal fuses to the individual components, a mechanical shield to protect the inlet system, and addition of a pressure sensor for the high vacuum mass spectrometer chamber.

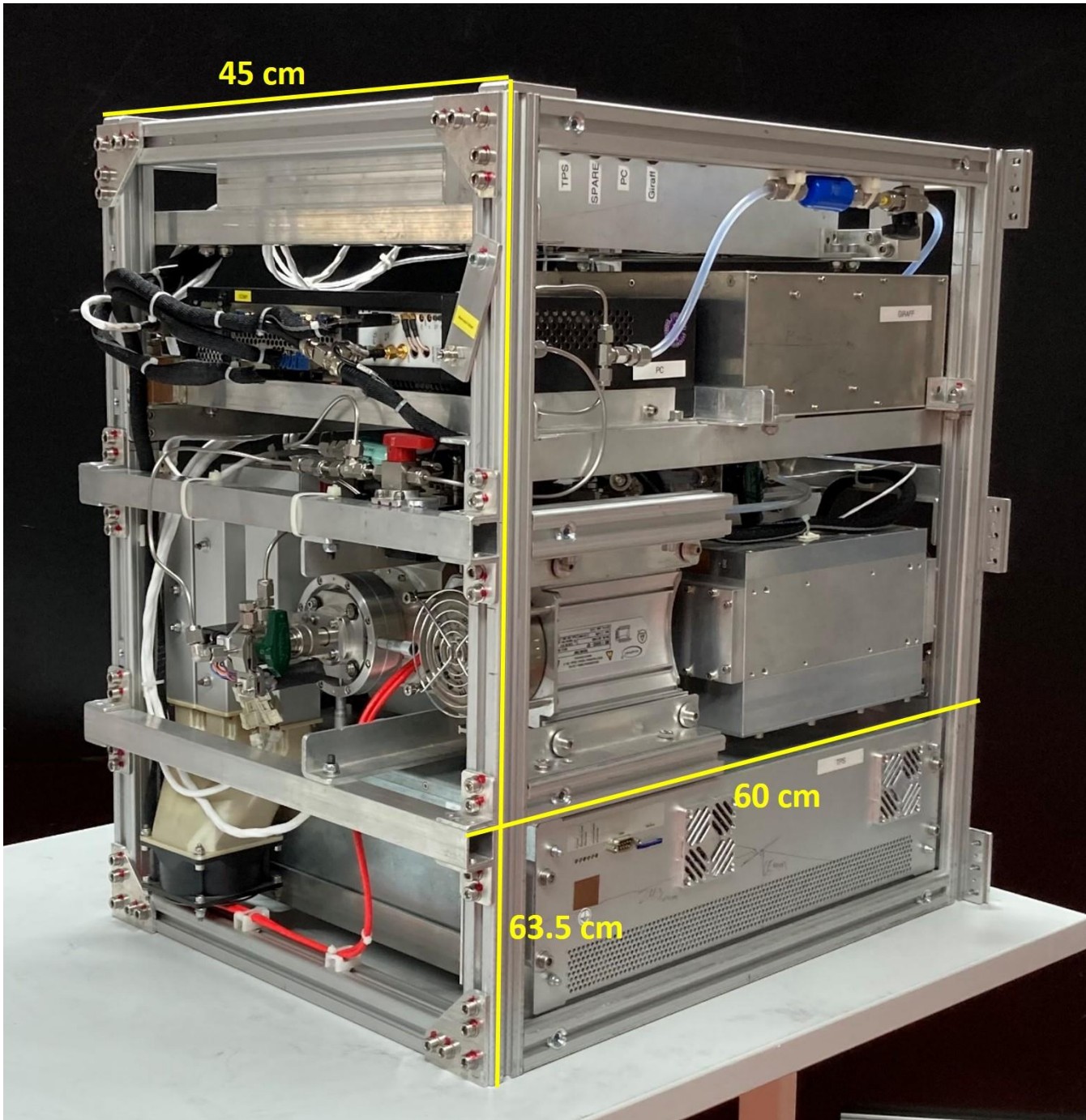

**Figure 2: The CARIBIC-AMS in its current (2024) configuration.**

### 2.2.3 Automated operation in the IAGOS-CARIBIC container

As no operators can be present in the cargo compartment to supervise instrument operation, all instruments have to run automatically and have to follow commands from the CARIBIC master computer that controls operation of the whole container-lab. Commands from the master computer are received regularly via an RS485 connection. Three operation states are set by the master computer: Initialization State (IN), Standby State (SB), or Measurement State (MS). The IN command urges the instrument to proceed to a state in which it consumes the minimal amount of power, while still being able to

commence the start-up process once the SB command is received. The instruments achieve the SB state when they are ready for starting their measurements. The master computer sends the MS signal to the different instruments if a defined ambient static pressure threshold is undercut. Typically, this value is 800 hPa when changing the signal from SB to MS (after take-off) and 870 hPa for changing from MS to SB (before landing). In the new setup in the A350, the weight-on-wheel signal will be used to start instrument operation.

This mode of operation differs significantly from the usual handling of any AMS, such that the CARIBIC-AMS had to be customized in hard- and software. A block diagram of the hardware and software components controlling the CARIBIC-AMS operation is shown in Figure A1. The new rack power distribution box (RPDB) contains a set of supercapacitors to provide sufficient power reserve (approx. 60 sec) to shut down the instrument in case of power cuts which are always to be expected onboard aircraft ground operation. A DC/DC converter transforms the incoming DC voltage from the container power supply

to 24 VDC for all components of the CARIBIC-AMS. Once the instrument receives the supply voltage, the capacitors are charged until full charge state is achieved. Only then the CARIBIC-AMS computer is started. Once the computer is started, the CARIBIC-AMS automatically goes into the IN state, i.e., the control software "VBus" and the automation software called "CARIBIC-MAN" start automatically. The VBus unit handles different hardware components, such as opening and closing of valves, and measures housekeeping data of the instrument. The CARIBIC-MAN software simulates a human operator to reach

the SB and MS states by executing predefined tasks. To secure the system against a computer crash, a watchdog is triggered every 180 seconds by the VBus software. If this trigger signal is not received within 180 seconds, the computer is re-booted. Upon receiving the SB command, the pre-pump (MD1, Vacuubrand, Germany) is switched on and runs at full speed. The valve between pre- and turbomolecular pump is opened (valve 1 in Fig. 1). The turbomolecular pump (SF270, Pfeiffer, Germany) is switched on with full power automatically after a vacuum of 5 hPa or less is reached in the vacuum chamber (P3

in Fig. 1). The pump speed, power, and pressure in the vacuum chamber are monitored constantly. Furthermore, the ToF Power Supply unit (TPS) is switched on. The VBus handles the opening of the different valves, the reading of the pressure sensors, regulation of the CPI, and the saving of the currents of the different units as RPDB, computer, pumps, etc. If good vacuum conditions ($p < 3 \times 10^{-6}$ hPa) are reached, the SB state is achieved. The CARIBIC-MAN sends then a message to the master computer that the CARIBIC-AMS is in SB. The time to reach the SB state is typically 10 min.

If the MS command is sent by the master computer, CARIBIC-MAN passes the command to open the inlet to the VBus. After that, the data acquisition software (DAQ) sets the voltages for the ion optics, the high voltages for the mass spectrometer, and

turns on the vaporizer. To start the ion source, the current through the filament is ramped up slowly until it reaches the predefined target value. After all voltages stabilized, the mass-to-charge (m/z) and single ion (SI) calibrations are executed. For more details on the calibration procedures see below (Sect. 2.3.2). Finally, the data acquisition is started and the full MS

state is reached. The time to reach MS from SB state is typically another 10 min. The data acquisition keeps running until a change is required, e.g. a SI calibration is scheduled, a SB command is received, or a power cut is recognized. At the end of a flight, the CARIBIC master PC sends the SB command, upon which CARIBIC-MAN initiates that the inlet valve is closed, the data acquisition is stopped and all voltages in the mass spectrometer are shut down. Finally, upon receiving the IN command, all components are switched off. When power is switched off, the PC of the CARIBIC-AMS is shut down with the

remaining power of the supercapacitors. An example of the relevant instrument parameters during a flight is shown in Fig. 3, for a flight sequence from Munich to Boston and back in November 2019. Before the first take-off at Munich, power was briefly available on ground (13:57 – 14:22), such that the master PC sent the SB command and the turbo pump started evacuating the vacuum chamber. Therefore, after take-off at 15:05, the MS pressure (red, second panel) reached its target value of $3\times10^{-6}$ hPa very soon, such that the inlet opened at 15:25 when the MS command was sent. The SB and MS operation states

are indicated by the gray shaded areas. Time periods with open inlet can be seen from the pressure in the ADL in the third panel (blue). The CPI keeps the ADL pressure constant at a predefined value (set to 1.8 hPa for this flight sequence). Mass spectrometer data were recorded between 15:44 and 20:07, the early ending being due to a data acquisition software error. Upon landing at Boston, the SB command was received, but shortly after power was cut abruptly, such that CARIBIC-MAN received "power bad information" (23:39), closed the inlet and switched off the system very quickly. The aircraft landed at

Boston at 23:39 and took off again at 01:49. The CARIBIC-AMS was powered again at 02:01. In this flight, the MS command was sent later, such that the instrument remained in SB state until the MS command was received at 03:29. The inlet was opened the measurements were started at 03:42 and ran until 07:56. At this point, CARIBIC-MAN received the SB command and switched off the instrument in a regular manner. After power was cut at about 08:24, the CARIBIC-MAN used the remaining power in the capacitors to keep the PC running and shut it down properly at 08:48. The aerosol data measured during

this flight are show in Sect. 3.1.

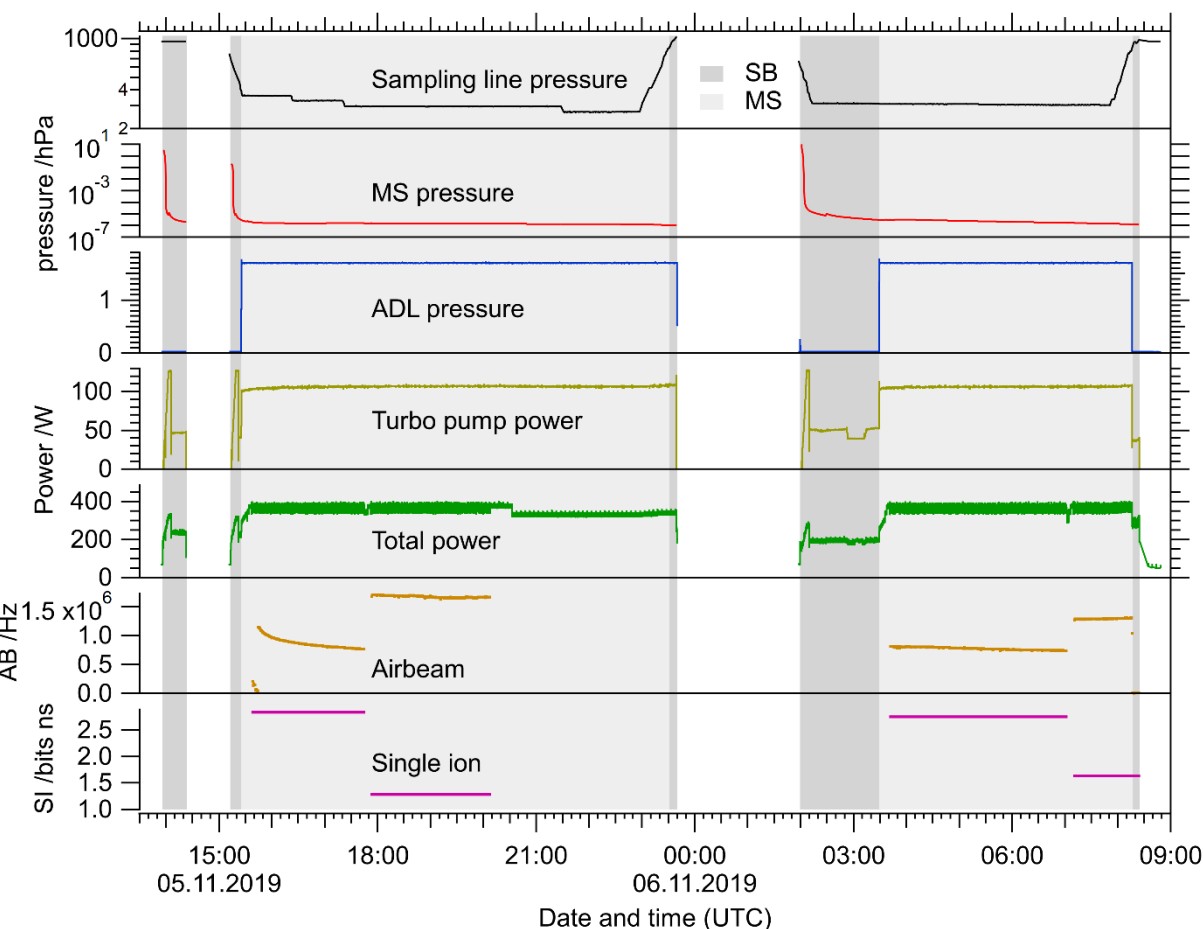

**Figure 3: Time series of housekeeping parameters of the CARIBIC-AMS recorded in November 2019, during flights 578 and 579 from Munich to Boston and back. The upper panel shows the sampling line pressure, the second panel shows the mass spectrometer chamber pressure (MS pressure), the third panel shows the pressure in the aerodynamic lens (ADL pressure). The fourth and fifth panels show the power consumed by the turbo pump and the total consumed power, respectively. The sixth panel shows the airbeam signal, the seventh the single ion signal. The operation states SB and MS are indicated by the shaded areas.**

All communications with the master computer as well as all tasks that are handled by CARIBIC-MAN successfully and all tasks that cannot be handled by CARIBIC-MAN due to erroneous conditions are monitored and archived in protocol files. CARIBIC-MAN checks whether necessary conditions for handling a task or reaching a state are fulfilled. If the conditions are not fulfilled, CARIBIC-MAN is able to detect these and tries to change the condition. This only works for known issues that are already implemented in the automation software code.

## 2.3 Instrument characterization

### 2.3.1 Inlet transmission of IAGOS-CARIBIC operation setup

The inlet used on the Airbus A340-600 is described in detail in Brenninkmeijer et al. (2007). The aerosol sampling lines inside the container were slightly modified when the CARIBIC-AMS was first integrated in 2017. The inlet sampling efficiency (Fig. 4 (a)) is estimated based on empirical equations from the literature (Baron and Willeke, 2001) and wind tunnel experiments with another aircraft-borne aerosol inlet (Hermann et al., 2001). The further transport efficiency through the sampling lines from the inlet to the container-laboratory (Fig. 4 (b)) and to the CARIBIC-AMS CPI (Fig. 4 (c)) were calculated using the

particle loss calculator (von der Weiden et al., 2009), for spherical particles with a density of 1.5 g cm$^{-3}$. Figure 4 (d) shows the total sampling line transmission from the outside air to the CPI of the CARIBIC-AMS. Also shown in Figure 4 (d) are three transmission functions of different ADL used in the AMS: The PM1 lens without CPI after Liu et al. (2007) and Knote et al. (2011), and the measurements at 250 hPa (UTLS conditions) by Molleker et al. (2020) using a CPI and a PM2.5 lens. The overall transmission from outside to the CPI of the CARIBIC-AMS is above 80% in a size range between 40 and 700 nm.

This corresponds well to the size ranges of the AMS inlet systems shown in Figure 4 (d). However, stratospheric aerosol particles may be larger (Brock et al., 2021), such that the upper cut-off of the inlet plays a role for the exact quantification of aerosol mass. For the new IAGOS-CARIBIC setup on the A350, a new inlet design has been developed which will have higher transmission properties for large particles. The influence of the transmission of the CPI together with the aerodynamic lens is discussed in the following section (2.3.2).


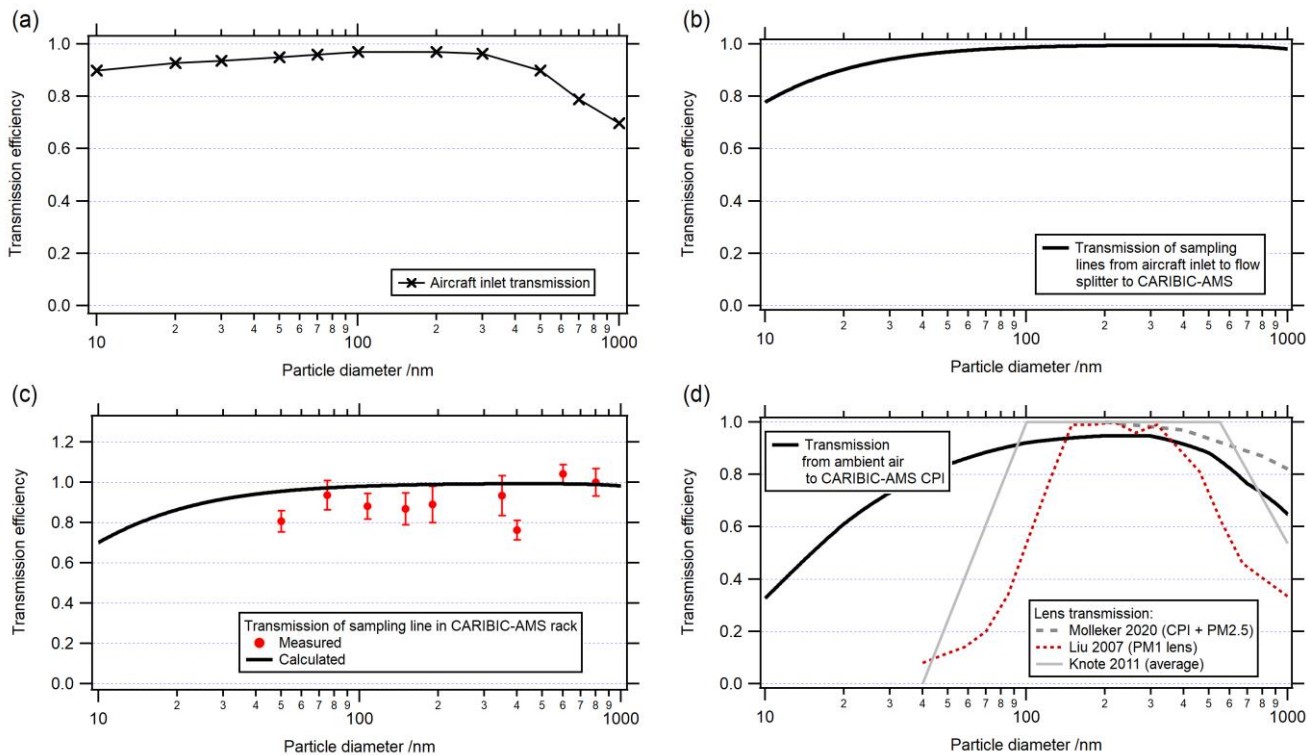

**Figure 4. Aerosol inlet transmission efficiency, derived by a combination of measurements and calculations: Transmission of the aircraft inlet (a), the sampling lines to the container-laboratory (b), the sampling lines in the CARIBIC-AMS rack (c), and total transport efficiency (d). Also shown in (d) are three transmission functions of the ADL, namely the PM1 lens without CPI after Liu et al. (2007) and Knote et al. (2011), and from Molleker et al. (2020) for 250 hPa (UTLS conditions), using a CPI and a PM2.5 lens.**

### 2.3.2 Calibrations

An aerosol mass spectrometer requires a set of calibrations that need to be performed regularly to maintain data quality: Inlet flow calibration, inlet transmission, particle size calibration, blank (gas-phase only) measurements, m/z calibration, single ion (SI) signal calibration, ionization efficiency (IE) calibration, and lens alignment.

In the CARIBIC-AMS, the CPI keeps the pressure in the aerodynamic lens constant. This means that the mass flow into the instrument is kept constant. Also, the acceleration of particles at the end of the aerodynamic lens which determines the particle velocity in the vacuum chamber is constant. Thus, flow calibrations and size calibration will not change, need to be calibrated only once, and later on only have to be checked. The pinched O-ring of the CPI however, should be changed from time to time due to mechanical deformation, and after replacing the inlet transmission has to be verified. How frequent these checks have to be performed will depend on the experience from the continued routine operation in future.

*In-flight calibrations*

A set of calibrations can be done in flight: m/z, SI, and blank measurement. After start-up of the filament and the vaporizer, nominally set to 600°C, CARIBIC-MAN turns on the data acquisition software and starts the m/z calibration window. Here,

three pre-selected m/z peaks (m/z 14, m/z 28, and m/z 44) are monitored and the m/z calibration is adjusted to these peaks. During acquisition, the m/z calibration is constantly updated, nominally every 5 seconds. Additionally, data analysis always allows for re-calibration of the mass spectra, as long as the raw signal is properly recorded. The SI calibration, which determines the signal level of single ions, can be done in pre-defined time intervals by CARIBIC-MAN. For this, the data acquisition software is stopped, the SI calibration window is opened and the calibration software procedure is run for 30

seconds. After that, the calibration results are transferred to the DAQ software, and data acquisition is restarted. The SI calibration has been conducted during the IAGOS-CARIBIC flights several times during a flight sequence. Figure 3 shows the recorded changes of the SI signal strength during the November 2019 flights, along with the resulting airbeam changes. The airbeam signal refers to the gas-phase signal at m/z 28 ($N_2^+$), which is used for monitoring the sensitivity of the instrument. Note that the airbeam signal decreases after a fresh pump-down, as can be seen in the data of the flight on November 05, 2019.

This is due to a background of low volatile organic compounds in the vacuum chamber which slowly evaporate after a fresh pump-down and produce ions such as $CO^+$ (on m/z 28, same as the $N_2^+$ signal) and $CO_2^+$. It may therefore be more reasonable to do the SI calibration only later in the flight. Blank measurements, which are required to determine the magnitude of gas-phase interferences on particle signals, are also conducted on a predefined time interval. For example, every three hours the 3-way valve (Valve 3 in Fig. 1) is turned such that the sampled air passes through the particle filter. After a set interval (nominally

10 minutes) the valve is turned again. Commands by CARIBIC-MAN and valve current along with the time stamp are monitored and logged.

*Laboratory calibrations*

Two types of calibrations cannot be conducted in flight but have to be done in the laboratory, namely the particle size calibration and the ionization efficiency (IE) calibration. The particle size calibration converts the measured particle velocity in the vacuum chamber into a vacuum aerodynamic particle diameter (DeCarlo et al., 2004). This requires test particles of

known size, shape, and density, e.g., PSL (polysterene latex) or ammonium nitrate ($NH_4NO_3$) particles. A particle size calibration with $NH_4NO_3$ particles for various ambient pressure values is shown in Fig. 5. As mentioned above, the CPI keeps the pressure in the aerodynamic lens constant, here the setpoint was 1.7 hPa. As a result, the particle velocity for a given vacuum aerodynamic diameter remains constant, such that the calibration fit function is independent on ambient pressure.

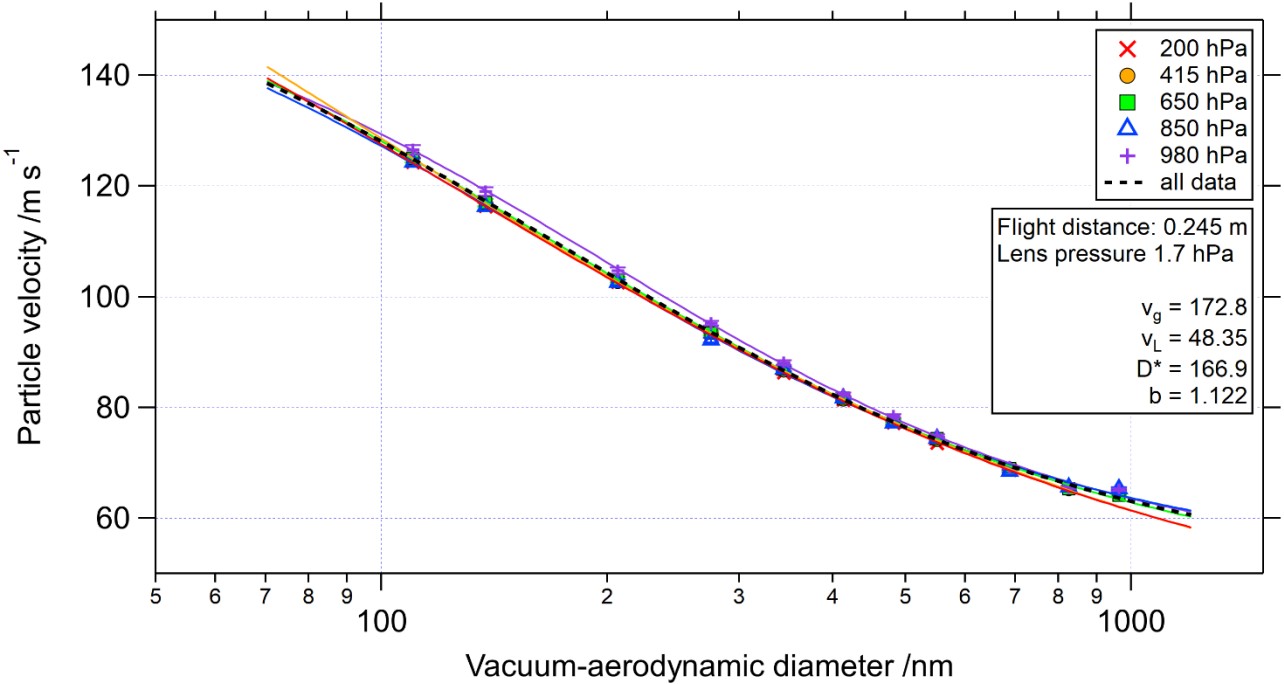

**Figure 5: Particle size calibration with NH₄NO₃ particles at different pressures. The constant pressure inlet (CPI) regulates the pressure inside the aerodynamic lens to a constant value, such that the size calibration remains stable over an ambient pressure range between 980 and 200 hPa.**

The IE calibration is regularly performed between the flight sequences. For this, we typically use the mass-based method as described in Drewnick et al. (2005) and Ng et al. (2011). Particles of known composition, size and number concentration are introduced into the instrument. The default calibration compound is ammonium nitrate. Particles are produced from an aqueous solution using an atomizer and a diffusion dryer. A differential mobility analyzer (DMA) is used to select the particle size. To minimize the contribution of multiply charged particles, the selected mobility diameter should be 300 nm or higher. The

particle concentration is monitored by a condensation particle counter (CPC) and converted into the reference mass concentration using the known size, density and shape factor of the particles. Then the ionization efficiency (ions per molecule) is determined by adjusting the CARIBIC-AMS response to the reference mass concentration. An example for a calibration in conducted in November 2018 for nitrate obtained from 300 nm ammonium nitrate particles is shown in Fig. 6. This calibration was done on ground level at approximately 1000 hPa.

We observed previously that the CPI used here does not have full transmission at low altitudes below 1500 m, because the opening of the O-ring loses its circular shape when it is squeezed too much. This had been documented already in Mei et al. (2020) by comparison between a C-ToF-AMS equipped with the CPI as used here and an HR-ToF-AMS using a CPI as described in Bahreini et al.(2008). For the CARIBIC-AMS application, this is only of minor relevance, since the main focus of the measurements is the UTLS region. However, for vertical profiles this effect needs to be corrected. For this purpose, the

mass-based IE calibration as described above is done at various pressures and a pressure-dependent IE is used for the data evaluation. This procedure assumes a size-independent transmission function of the CPI-ADL combination which represents a simplification. A size-resolved transmission measurement of the current CPI-ADL combination has not been done so far, but we assume that it is similar to the results reported by Molleker et al (2020). In-flight comparison with collocated measurements has not been possible during the IAGOS-CARIBIC flights yet, but comparisons during the TPEx campaign in 2024 with

volume concentrations inferred from UHSAS data (see section 3.2 and Joppe et al., 2025) show satisfying agreement between both instruments for altitudes above about 8000 m.

Relative ionization efficiency (RIE) values for ammonium and sulfate are calculated by adjusting the ammonium mass concentration during calibration according to the molar mass ratio of ammonium nitrate, and by using ammonium sulfate to obtain the RIE of sulfate. For organics, a RIE of 1.4 (± 0.3) is typically used for ambient aerosol (Xu et al., 2018).


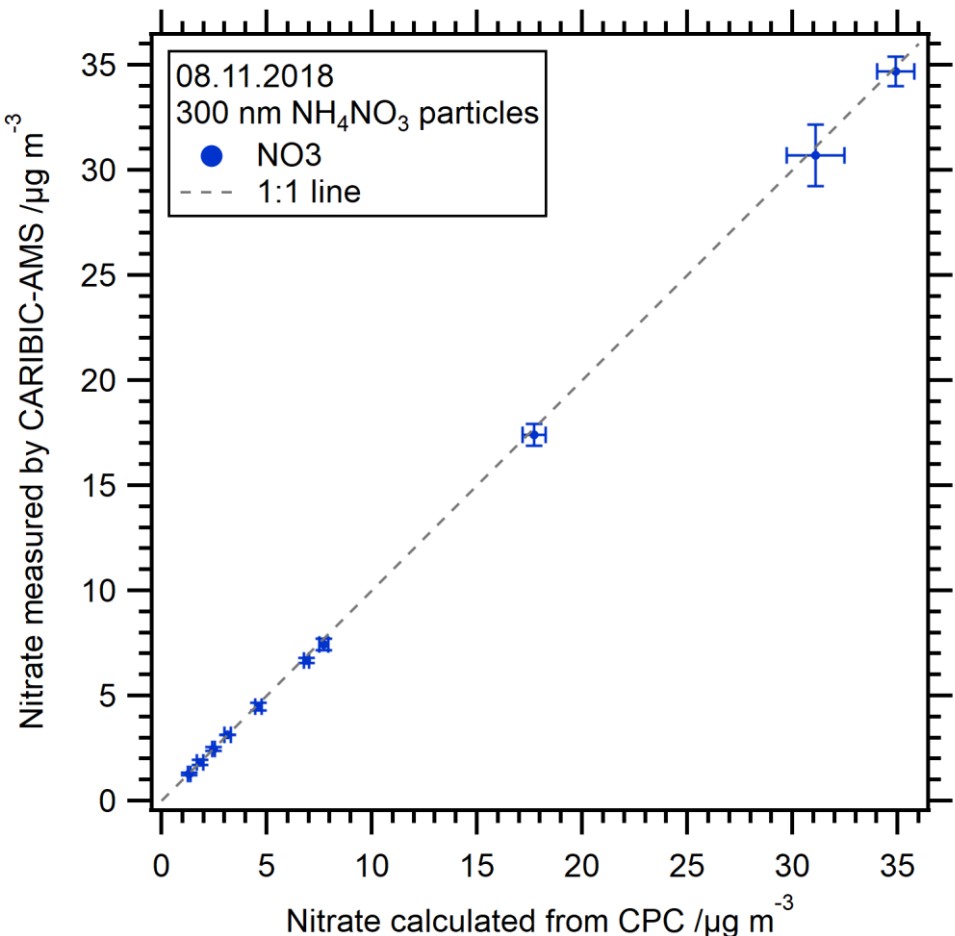

**Figure 6: Ionization efficiency calibration: The reported nitrate mass concentration from CARIBIC-AMS is scaled to the nitrate mass concentration calculated from particle number concentration during calibration with NH₄NO₃ particles of 300 nm. The**

ionization efficiency was adjusted such that the slope of the linear fit equals unity. **The resulting IE value is $3.12 \times 10^{-8}$ ions molecule$^{-1}$.**

**Error bars denote the standard deviations during the averaged measurement times.**

To ensure data quality and traceability over time, the IE calibration is conducted between each flight sequence at the laboratory of the Max Planck Institute for Chemistry (MPIC), Germany. To avoid unnecessary transport of the instrument between KIT and MPIC it is currently evaluated whether it will be possible in future to conduct the IE calibration between flight sequences at KIT.

**2.3.2 Intercomparison between the CARIBIC-AMS and other AMS instruments**

For further quality assurance, we conducted comparisons of the CARIBIC-AMS with two other aerosol mass spectrometers, namely a HR-ToF-AMS owned and operated by TROPOS, Leipzig, Germany, and a C-ToF-AMS owned and operated by MPIC. The C-ToF-AMS of MPIC is also used for aircraft measurements and is therefore also equipped with the CPI (Schulz et al., 2018). Comparison with the HR-ToF-AMS was conducted in October and November 2018 at the laboratory at TROPOS,

Leipzig. Both instruments were calibrated simultaneously with the same test aerosol (see Fig. 6 for the CARIBIC-AMS calibration). Figure 7 (a) shows the obtained mass concentrations of nitrate from both instruments. Both instruments respond linearly to particulate mass concentrations between 0 and 30 $\mu$g m$^{-3}$. Above 30 $\mu$g m$^{-3}$, small deviations from the 1:1 line occur, but such high concentrations are not relevant for UTLS conditions. The  uncertainty for the quantification of nitrate mass concentrations of 10% (Bahreini et al., 2009) is shown by the shaded area.

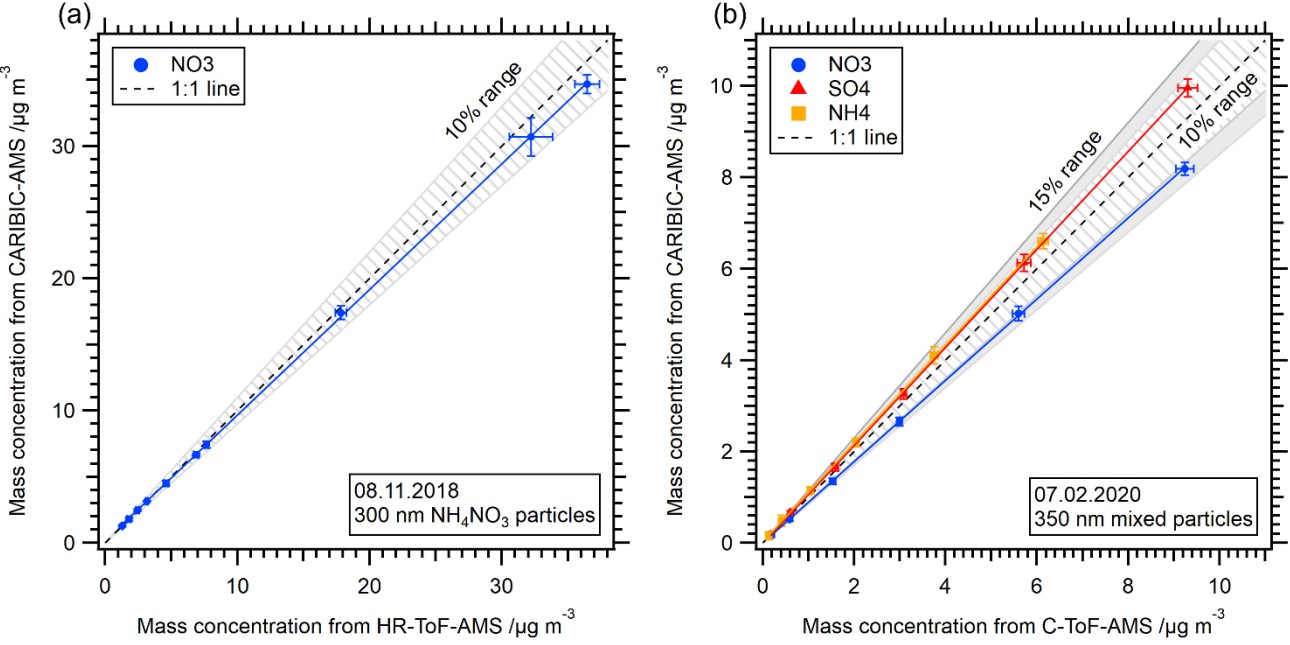


**Figure 7: (a) Comparison between nitrate mass concentrations measured by the CARIBIC-AMS and a HR-ToF-AMS for laboratory-generated ammonium nitrate particles of 300 nm diameter. (b) Comparison between the CARIBIC-AMS and C-ToF-AMS with mixed 350 nm particles containing ammonium nitrate and ammonium sulfate. Both instruments were calibrated with 350 nm ammonium nitrate particles. Sulfate and ammonium were calculated independently for both instruments using internal**

 **calibration (relative ionization efficiency) with pure NH$_4$NO$_3$ and (NH$_4$)$_2$SO$_4$ particles. Error bars denote standard deviations during the averaged time periods. The 10% uncertainty range refers to the nitrate comparison, the 15% range to the RIE uncertainty (ammonium and sulfate comparison) (Bahreini et al., 2009).**

The comparison between the CARIBIC-AMS and the C-ToF-AMS (Fig. 7 (b)) took place in February 2020 at the laboratory of MPIC, Mainz. Both instruments sampled simultaneously from the same particle generation setup. For calibration, 350 nm

ammonium nitrate particles were used. For both instruments, the IE value was determined by applying the CPC-based method as described above. The RIE values for ammonium and sulfate were obtained independently for both instruments by internal calibration with ammonium nitrate and ammonium sulfate (C-ToF-AMS: RIE$_{NH4}$ = 3.47; RIE$_{SO4}$ = 1.51; CARIBIC-AMS: RIE$_{NH4}$ = 4.07; RIE$_{SO4}$ = 1.01). For the comparison between both instruments, we used internally mixed 350 nm particles composed of ammonium sulfate and ammonium nitrate. Although both instruments were calibrated on the same day of the

comparison, the data in Fig. 7 (b) show deviations from the 1:1-line. In mixed particles, the individual compounds appear to have slightly different relative ionization efficiencies than they have in pure particles. However, the deviations are much less than the typically reported uncertainty of 15% for AMS RIE measurements (Bahreini et al., 2009) which is denoted by the gray area in Fig. 7 (b).

### 2.3.3 Size-resolved composition measurements

The CARIBIC-AMS is equipped with a pseudo-random multi-slit chopper wheel (see Fig. 1), termed ePToF (efficient particle time-of-flight) chopper (Goetz et al., 2018; Saarikoski et al., 2019). In contrast to the conventional single-slit chopper wheel that has a duty cycle of about 2%, the ePToF chopper has a duty cycle of 50%, thereby allowing for a more efficient measurement of size-resolved particle mass concentration. This is especially important for aircraft measurements, where particle mass concentrations are low and a high time resolution is required.

The original electronics box that regulates the automation of the chopper and its synchronization with the mass spectrometer needed to be removed due to space limitations in the new rack. The hardware of the custom-made automation of the ePToF-chopper is now integrated in the VBus unit. The chopper itself is controlled via a microprocessor with frequency and phase automated rotating field. The microprocessor gets feedback of the rotation via an optical scanning of the chopper wheel. Once per rotation the microprocessor gets a pulse from the mass spectrometer. For accurate ePToF measurements, the chopper

rotation has to be synchronized with the pulsed ion extraction in the mass spectrometer. In the current configuration, the mass spectrometer is pulsed with 81.6 kHz such that more than 700 mass spectra are recorded per chopper cycle. The synchronization is achieved as follows: the microprocessor generates a virtual signal which is in phase with the extraction pulses generated by the mass spectrometer. The real phase of the chopper is adjusted relative to the virtual signal. By this, the percentage of complete chopper rotations that can be used for measurements is above 95 %.

Figure 8 shows a comparison between measurements taken with the normal PToF mode (single slit with duty cycle of 2%) and the ePToF mode (50% duty cycle). The data were taken in the laboratory of MPIC with the CARIBIC-AMS. The chopper wheel installed in the CARIBIC-AMS allows for both operation modes, such that the measurements were taken directly after

each other, using $NH_4NO_3$ particles of 300 nm diameter with a concentration of 110 cm$^{-3}$. The sampling line pressure was 300 hPa to simulate in-flight conditions. Shown is the signal at m/z 46 versus the flight time of the particles between chopper and vaporizer, averaged over 8 seconds. The particles arrive after 2400 µs at the vaporizer and the maximum of the signal is at about 2500 µs (2545 µs for PToF and 2555 µs for ePToF). The insert shows that the ePToF signal is slightly narrower than the PToF signal. A Gauss fit to the peaks yields a width of 106 µs for PToF and 66 µs for ePToF.

The size distribution mode has not been used during aircraft operation yet. During the IAGOS-CARIBIC flights between 2018 and 2020, the development of the electronic synchronization had not been completed. For TPEx, it was decided to focus on mass concentrations only to obtain a higher signal-to-noise ratio together with the best possible time resolution. This was required to meet the objectives of the campaign which focused on small-scale mixing processes in the tropopause region.

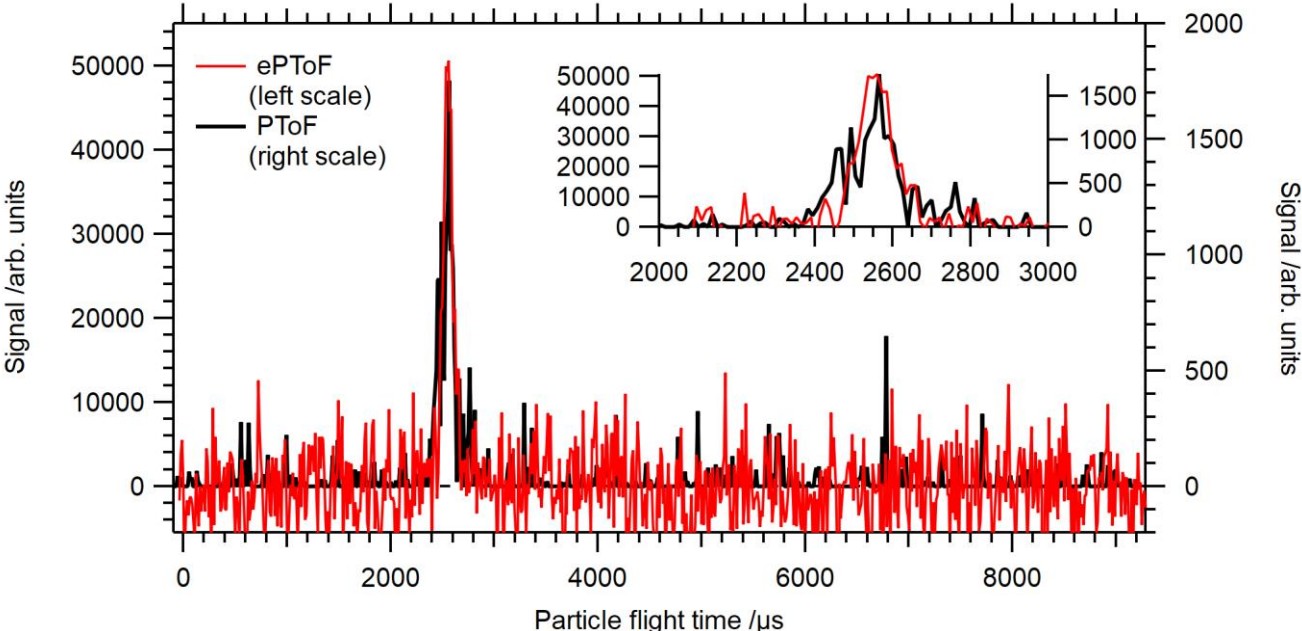

**Figure 8: Comparison between normal PToF and ePToF size distributions, measured in the laboratory with 300 nm $NH_4NO_3$ particles (300 hPa, particle concentration 110 cm$^{-3}$). The insert shows an enlargement of the particle signal, ranging over about 200 µs. The ePToF signal is slightly narrower than the PToF signal.**

## 3. Aircraft application

The first aircraft operation of the CARIBIC-AMS during an IAGOS-CARIBIC flight sequence took place in May 2018. Until the phase-out of the A340 by Lufthansa in March 2020, the CARIBIC-AMS was operated during 11 flight sequences corresponding to 46 flights (one of these sequences contained 6 flights). Not all flights were successful due to a variety of hardware and software related teething problems that occurred during the fully automated flight operation. During the operational break due to the changeover from the A340 to the A350, the CARIBIC-AMS was operated on a Learjet 35A in the

TPEx campaign in June 2024 (Bozem et al., 2025; Joppe et al., 2025). Here we briefly report results from these aircraft applications to demonstrate the instrumental capabilities of the CARIBIC-AMS.

### 410 3.1 IAGOS-CARIBIC Flights

Figure 9 shows an overview of all flights during which the CARIBIC-AMS was operated on IAGOS-CARIBIC. Black lines denote all flight tracks, red dots denote flight periods during which CARIBIC-AMS data were recorded. It must be noted that not all flights during which data were recorded yielded useful data. Several instrumental issues occurred during these first fully automated flights, such that only a subset of the flights can be used for further analysis.

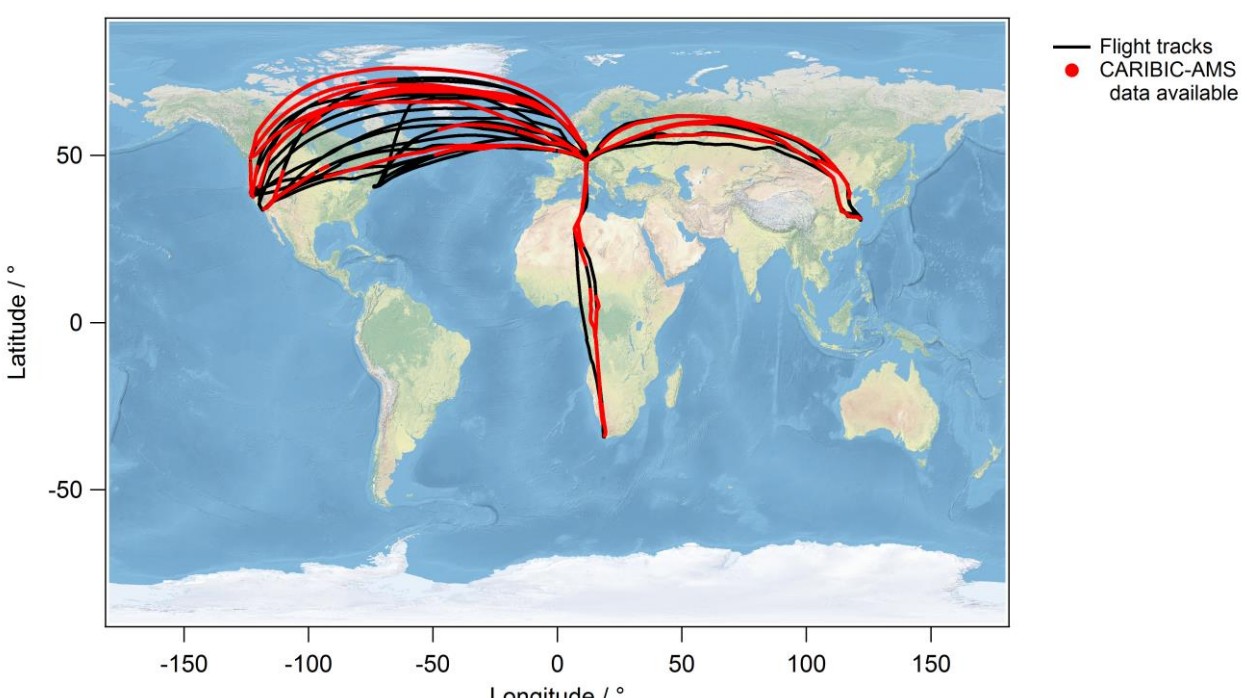


**Figure 9: Map of IAGOS-CARIBIC flights for which the CARIBIC-AMS was installed in the container. Red dots denote flight segments during which CARIBIC-AMS data were recorded during the time period between May 2018 and March 2020.**

As an example, we present here data of the IAGOS-CARIBIC flights 578 and 579, corresponding to Fig. 3. Both flights are part of the flight series performed in November 2019. Flight 578 took place on Nov 5, 2019, from Munich to Boston, the return

flight 579 took place on the next day. Figure 10 (a) shows the time series of flight altitude, ozone ($O_3$) and carbon monoxide (CO) (Zahn et al., 2024), along with the particulate sulfate mass concentration measured by the CARIBIC-AMS. The sulfate mass concentration is shown with the original 30 s time resolution and additionally with 5 min resolution, each data point being the average over 10 original data points. For the data evaluation, we used an operationally defined constant collection efficiency (CE) of 0.5. This is a first-guess simplification instead of using the composition-dependent CE (CDCE)

recommended by Middlebrook et al. (2012). However, the high detection limits for ammonium impose a problem for determining the correct CDCE, especially under acidic conditions as in the stratosphere, because for this a good ammonium measurement is required. Thus, we likely overestimate sulfate in the stratosphere. On the other hand, the limited particle transmission of the inlet system for particles larger than 800 nm (see Fig. 4) likely leads to an underestimation of stratospheric sulfate, because, as shown in Brock et al. (2021), stratospheric particle size distributions of sulfate may extend to 1 μm and

above.

In the first flight, the mass spectrometer measurement started about 45 min after take-off. Power had been available for about 30 min shortly before take-off (see Sect. 2.2.3 and Fig. 3), such that pumping of the vacuum chamber had started already on ground and the required vacuum chamber pressure was reached rather quickly in flight. In the second flight, the MS command was sent only at 03:29 (see Fig. 3) such that the measurements could start only at this time. In the first flight, the measurements

stopped around 20:00 due to a data acquisition software error.

The calculated detection limits (DL, see Sect. 3.3) for the 30 s resolution sulfate mass concentration are also given in Fig. 10. Most of the sulfate data points are above the detection limit. In general, the detection limit is higher in the beginning than at the end of the flight. The reason for this is the limited pumping time before starting the measurements, such that the background signal in the vacuum chamber decreases over time during a flight.

During both flights, the measured sulfate mass concentration follows the $O_3$ mixing ratio. This finding is typical for the lower stratosphere, as has been observed previously in several studies (e.g., Martinsson et al., 2009; Andersson et al., 2013; Joppe et al., 2024). To closer inspect the transition region between troposphere and stratosphere during these two flights, Fig. 10 (b) shows the tracer-tracer correlation between $O_3$ and CO, which is a typical metric for mixing across the tropopause (Fischer et al., 2000; Hoor et al., 2002; Schmale et al., 2010; Joppe et al., 2024). The grey data points represent all $O_3$ and CO data, the

colored data points those times when also sulfate was measured, color-coded by the sulfate mass concentrations. It can clearly be seen that the highest sulfate concentrations are measured at the highest $O_3$ and lowest CO levels. The data points with low $O_3$ (around 60 ppb) and high CO (above about 70 ppb) indicate tropospheric air. The chemical tropopause, representing the transition from tropospheric air to the mixing layer between troposphere and stratosphere (Hoor et al., 2002; Zahn and Brenninkmeijer, 2003; Pan et al., 2004), can be estimated here to be at around 70 ppb $O_3$. Thus, almost all data recorded by

the CARIBIC-AMS during this flight sequence were taken in the mixing layer.

In comparison to previous IAGOS-CARIBIC data from filter samples with offline analysis as mentioned above (Andersson et al., 2013, 2015; Martinsson et al., 2014), our sulfate data lie in a similar range. The sulfur-to-ozone ratio from the time period 2005 – 2008, sampled before the eruption of Kasatochi and described by Andersson et al. (2013) as "somewhat volcanically influenced samples", is about 0.33 ng m$^{-3}$ STP/ppbv. STP stands for Standard Temperature (273.15 K or 0°C) and Pressure

(1000 hPa) (IUPAC, 2025). Applying this ratio to our data and converting sulfur to sulfate, we find very similar values for the stratospheric part of our data. Compared to measurements conducted onboard the research aircraft HALO in May and June 2020 over central Europe and the North Atlantic (Joppe et al., 2024), the stratospheric sulfate mass concentrations were higher in November 2019. Especially during flight 578 on Nov. 05, an air mass with sulfate mass concentrations between 0.3 and 0.5

µg m⁻³ STP sulfate was encountered between 18:00 and 20:00, at $O_3$ levels of about 300-400 ppb. Such values were not observed in May and June 2020. This may reflect the influence of the eruption of the volcano Raikoke in June 2019 that injected about 1.5 Tg of sulfur dioxide into the stratosphere (Vernier et al., 2024; Leeuw et al., 2021; Osborne et al., 2022) and caused enhanced aerosol in the stratosphere until November 2019 and beyond (Kloss et al., 2021; Boone et al., 2022).

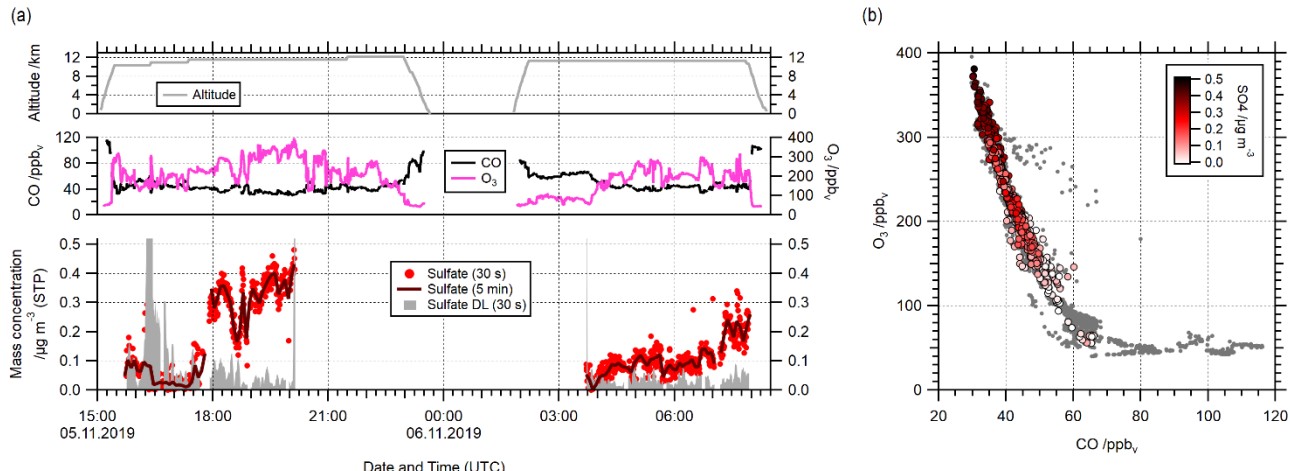

**Figure 10: (a) Time series of $O_3$, CO, and particulate sulfate from IAGOS-CARIBIC flights FL578 and FL579, conducted on November 05 and 06, 2019, between Munich and Boston. (b) Tracer-tracer plot with $O_3$ and CO, color-coded by sulfate mass concentration.**

## 3.2 TPEx 2024 example flight

During the TPEx campaign in 2024, eight research flights with a Learjet 35A were conducted over the North Sea, the Baltic Sea, Germany, Denmark, and Sweden (Bozem et al., 2025; Joppe et al., 2025). The CARIBIC-AMS was operated during all measurement flights. In contrast to the fully automated operation in the IAGOS-CARIBIC container, the CARIBIC-AMS was operated manually during TPEx, with about 3 hours pumping time before take-off.

Here, we present data from one example flight on June 12, 2024. The flight took place over Northern Germany, Denmark, and Sweden, between 54 and 59°N, and 9.5 and 13.5°E, at altitudes up to 12 km. Figure 11 (a) shows the sulfate mass concentration along with flight altitude, ozone and CO. Again, a constant collection efficiency of 0.5 was used for the calculation of the aerosol mass loading. Sulfate mass concentrations are shown for ambient pressure below 500 hPa, ozone and CO data refer to altitudes above 2 km altitude. Ozone mixing ratios were measured using the 2BTech Model 205 instruments (Johnson et al., 2014), CO was measured using the University Mainz QCL infrared absorption spectrometer (UMAQS, Müller et al., 2015).

Although the flight did not reach higher than 12 km, the ozone mixing ratio shows values of up to more than 900 ppbv, indicating that the Learjet reached deep into the lower stratosphere. At 300 - 400 ppb $O_3$, the particulate sulfate mass concentration is about 0.5 - 0.6 µg m⁻³, comparable to the values measured during the IAGOS-CARIBIC flight 578 on Nov

05, 2019. However, the sulfate mass concentration reaches a maximum of about 0.7 µg m$^{-3}$ at O$_3$ mixing ratios of 600 ppbv (12:10 - 12:40), and decreases down to 0.5 µg m$^{-3}$ at the highest O$_3$ levels of 900 ppbv. Figure 11 (b) shows the correlation between O$_3$ and CO, color-coded by sulfate mass concentration. According to the season and the northern latitude, the ExTL

spans a broader range of CO and O$_3$ mixing ratios compared to the IAGOS-CARIBIC flights shown in Fig. 9. Ozone increases already at 100 ppbv CO, while for IAGOS-CARIBIC (Fig. 10 (b)), it remained at its tropospheric values of about 60 ppb until CO decreased to 70 ppbv. Also from this graph, it can be seen that the sulfate maximum (darkest colors) is found in this flight between 500 and 600 ppbv O$_3$ and around 40 ppbv CO.

Also shown in the lowest panel of Fig. 11 (a) is a comparison between the CARIBIC-AMS and data from a UHSAS (Ultra-

high sensitivity aerosol spectrometer) which was also operated onboard the Learjet (Joppe et al, 2025) during TPEx. For the comparison, both data sets were averaged over 10 minutes and converted to total volume concentration, assuming spherical particles and an average particle density of 1.5 g cm$^{-3}$. The agreement is satisfying for altitudes above 8 km, but the noisy signals of organics and ammonium lead to the large scatter of the CARIBIC-AMS data points. As mentioned above, the transmission efficiency of the CPI – ADL combination likely leads to the underestimation by the CARIBIC-AMS at lower

altitudes. Further examples of the CARIBIC-AMS – UHSAS comparison during TPEx are given in Joppe et al. (2025).

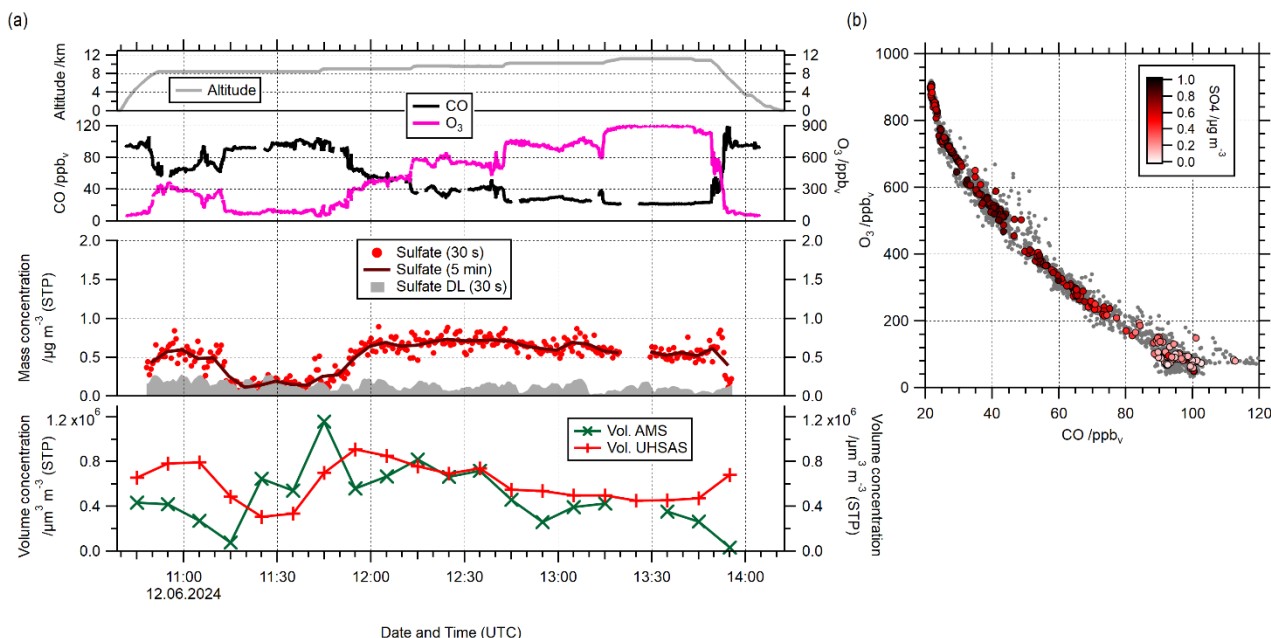

**Figure 11: (a) Time series of O$_3$, CO, and particulate sulfate from research flight 4 of the TPEx campaign (June 12, 2024). Shown are the raw data (30 s), a 5-min average, and the detection limit for the 30 s resolution. The lowest panel shows a comparison between**
**the total volume concentration measured by the CARIBIC-AMS and a UHSAS operated in parallel on the Learjet. (b) Tracer-tracer plot with O$_3$ and CO, color-coded by sulfate mass concentration.**

## 3.3 Detection limits

Detection limits are determined routinely from in-flight data following the method described in Schulz et al. (2018). This method uses the short-term variability of the background signal in the instrument which is measured throughout all data acquisition by the CARIBIC-AMS. Thus, the detection limits can be determined for each flight situation. It can be seen in Fig. 10 (a) that the detection limit (here for sulfate) varies due to short-term variability of the background signal, especially during the first half of the flight. Detection limits can also be calculated from the automated blank filters as $3 \times \sigma_{Filter}$ ($\sigma$: standard

deviation of the calculated mass concentration during the blank measurement; e.g., Drewnick et al., 2009). Figure C1 shows a comparison of both methods for flight 578. The automated blanks were taken every hour for 10 minutes during this flight. In the beginning of the flight, the background signal method yields higher detection limits than the blank filter method, but after about three hours, both methods agree fairly well. To calculate a representative DL for this flight, we chose a period of flight 578 where the background signal was smooth (18:40 - 20:00) and determined the average DL for this flight period. These

values are given in Table 1, along with calculated detection limits for longer averaging times (5 min and 30 min). It has to be emphasized that the in-flight values are valid only for the given time period of flight 578 and refer to a CE of 0.5 that was used for the data evaluation here. The determined 30-second DL for sulfate with 0.035 µg m$^{-3}$ STP is comparable to the value of 0.03 µg m$^{-3}$ STP reported in Schmale et al. (2010) and Schulz et al. (2018) for airborne C-ToF-AMS operation. Also, the chloride DL with 0.022 µg m$^{-3}$ STP is similar to that reported in Schmale et al. (2010). For the other substances, the DL values

determined in flight 578 are higher by a factor of 3 (nitrate), 4 (ammonium) and 6 (organics) compared to the reported DL values for the C-ToF-AMS. In the laboratory, lower DL values are achieved. Data measured in August 2024 with 10 seconds time resolution are also included in Table 1. These values are comparable to the in-flight values for nitrate, chloride and ammonium, but better for sulfate and organics. During the TPEx campaign, detection limits were higher, likely due to the short pumping time before the flights and the short flight duration of about 3-4 hours. Only on flight days where two flights after

each other were conducted, the conditions for reaching good vacuum conditions were slightly better. The calculated detection limits for this flight (F08 on 17 June 2024) are included in Table 1. However, we are confident that the ongoing instrumental improvements and further developments will ensure that these values can be surpassed in future application. One possible solution is filling the instrument with nitrogen or synthetic air after lab calibration before integration into the container-laboratory before each flight sequence.


**Table 1: Detection limits in µg m$^{-3}$ STP for the original in-flight time resolution of 30 seconds as well as for two examples for longer averaging times, inferred from flight 578 (18:40 - 20:00). Also given are 10-second values measured in the laboratory in August 2024, and 30-second values calculated from the 10-second laboratory data. The values refer to a collection efficiency of 0.5 which was used for the data evaluation.**

| | In-flight IAGOS-CARIBIC | | | In-flight TPEx F08 | Laboratory | |
| --- | --- | --- | --- | --- | --- | --- |
| | 30 sec | 5 min | 30 min | 30 sec | 10 sec | 30 sec |
| Organics | 0.69 | 0.22 | 0.089 | 1.56 | 0.25 | 0.14 |

|  | In-flight IAGOS-CARIBIC | | | In-flight TPEx F08 | Laboratory | |
| --- | --- | --- | --- | --- | --- | --- |
|  | 30 sec | 5 min | 30 min | 30 sec | 10 sec | 30 sec |
| Nitrate | 0.055 | 0.017 | 0.007 | 0.16 | 0.056 | 0.032 |
| Sulfate | 0.035 | 0.011 | 0.005 | 0.10 | 0.025 | 0.014 |
| Ammonium | 0.38 | 0.12 | 0.049 | 1.31 | 0.35 | 0.20 |
| Chloride | 0.022 | 0.007 | 0.003 | 0.10 | 0.019 | 0.011 |


## 4. Summary and Conclusion

We presented a fully automated and compact aerosol mass spectrometer (CARIBIC-AMS) for operation during regular monthly flights in the UTLS as part of the IAGOS-CARIBIC payload. The instrument is in operation since May 2018 and had

its first fully successful scientific flight in October 2018. The total number of flights conducted until March 2020 is 46.

The original instrument, a commercial mAMS by Aerodyne Research Inc., was redesigned to match the requirements of fully autonomous operation inside the IAGOS-CARIBIC container-lab and aviation safety. This modification included mechanical reconstruction of the rack to fit into the available space in the container, aircraft safety measures like reduction of flammable material and installation of thermal fuses, and installation of automatic valves to allow for computer-controlled operation.

For the operation, an automation software was written that performs the required steps that are usually done by the user. The software monitors all relevant instrument parameters and follows the commands given by the master computer inside the IAGOS-CARIBIC container. Depending on these commands, the CARIBIC-AMS remains in a standby-by state or proceeds to measurement mode. Several in-flight calibration routines are regularly performed by the control software. Power cuts are backed up by supercapacitors that provide sufficient power to safely shut down the instrument.

Now that the CARIBIC-AMS is ready for regular operation, it will provide a unique dataset on aerosol composition in the UTLS, an atmospheric region of which our understanding is still limited by a lack of in-situ data. The time resolution of 30 seconds allows for detection of spatial structures on the order of 7000 m. For lower detection limits a reanalysis of the data after flight using lower time resolution is always possible. Flights will be conducted on a regular basis to various destinations on the northern but also the southern hemisphere.

Here, only a first data example was presented to demonstrate the capabilities of the CARIBIC-AMS. In future, a more detailed data analysis will include also the full set of atmospheric composition parameters that is measured by the instrumentation inside the IAGOS-CARIBIC container. Aerosol data include total number concentration for different lower cut-off diameters, particle size distribution, as well as concentration of non-volatile particles, black carbon, and biological particles. Gas phase tracers like $O_3$, CO, $NO_y$, acetonitrile and many others provide information on air mass origin and particle sources such as

industrial emissions, aircraft exhaust, or biomass burning. Meteorological reanalysis (ERA-5) and chemistry-transport models will also be used to further analyze the data.

Currently the transfer of the IAGOS-CARIBIC project to a new Lufthansa aircraft, an Airbus A350, is in progress. IAGOS is a European Research Infrastructure for global observations of atmospheric composition from commercial aircraft (www.iagos.org) such that a long-term operation of the IAGOS-CARIBIC system to obtain a globally representative UTLS

dataset is secured.

## Appendix A

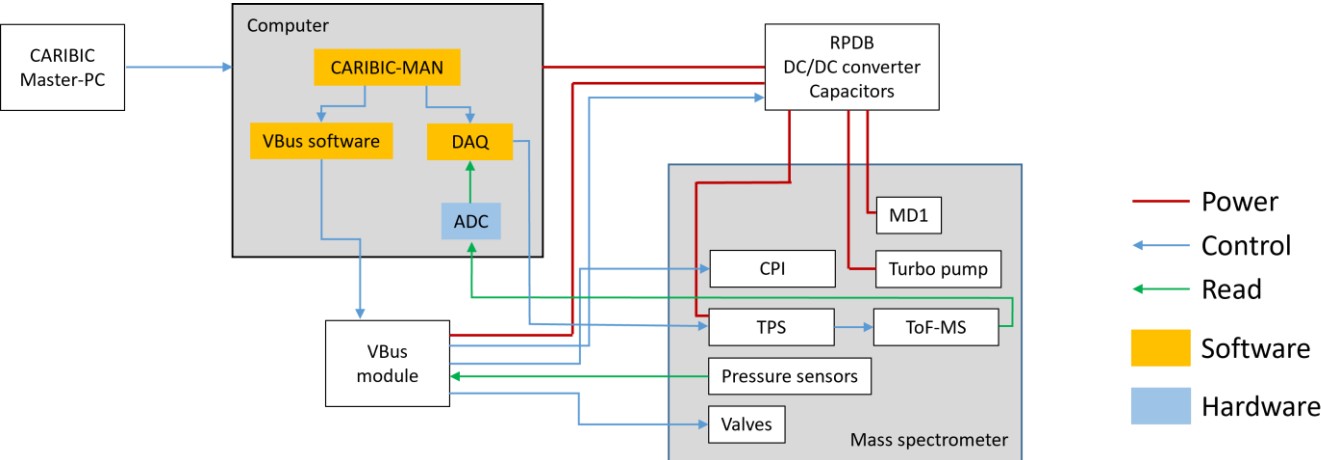

**Figure A1: Schematic representation of the software and hardware hierarchy of the CARIBIC-AMS: The main control software**
**running on the computer is CARIBIC-MAN, while the VBus software is controlling the VBus module which in turn controls the hardware components such as the RPDB (Rack Power Distribution Box), the valves and the CPI. CARIBIC-MAN also controls the data acquisition software (DAQ) that controls the ToF power supply (TPS) and reads the data through the analog-digital converter (ADC).**


## Appendix B

**Table B1: Overview on ionization efficiency (IE) measurements during the IAGOS-CARIBIC operation time of the CARIBIC-AMS (October 2018 until March 2020), during laboratory measurements in November 2020, and after the TPEx campaign in June 2024.**
**The data are averaged values from measurements taken at pressures between 200 and 400 hPa, representative for the UTLS region. To minimize the effect of multiple charged particles, only particles with 350 and 400 nm diameter have been used. AB = air beam (ion rate of $N_2^+$ ions). The ratio IE/AB is an indicator for sensitivity changes of the instrument.**

| Date | IE (ions molecule$^{-1}$) | AB (s$^{-1}$) | IE/AB (ions (molecule s)$^{-1}$) |
|---|---|---|---|
| June 2019 | 8.44e-8 | 1.88e6 | 5.08e-14 |
| August 2019 | 1.07e-7 | 1.93e6 | 5.38e-14 |
| October 2019 | 1.73e-7 | 2.06e6 | 8.39e-14 |
| November 2019 | 2.01e-7 | 1.94e6 | 1.04e-13 |
| November 2020 | 4.40e-8 | 1.76e6 | 2.50e-14 |
| July/Aug 2024 | 6.23e-8 | 4.09e5 | 1.52e-13 |

**Appendix C**

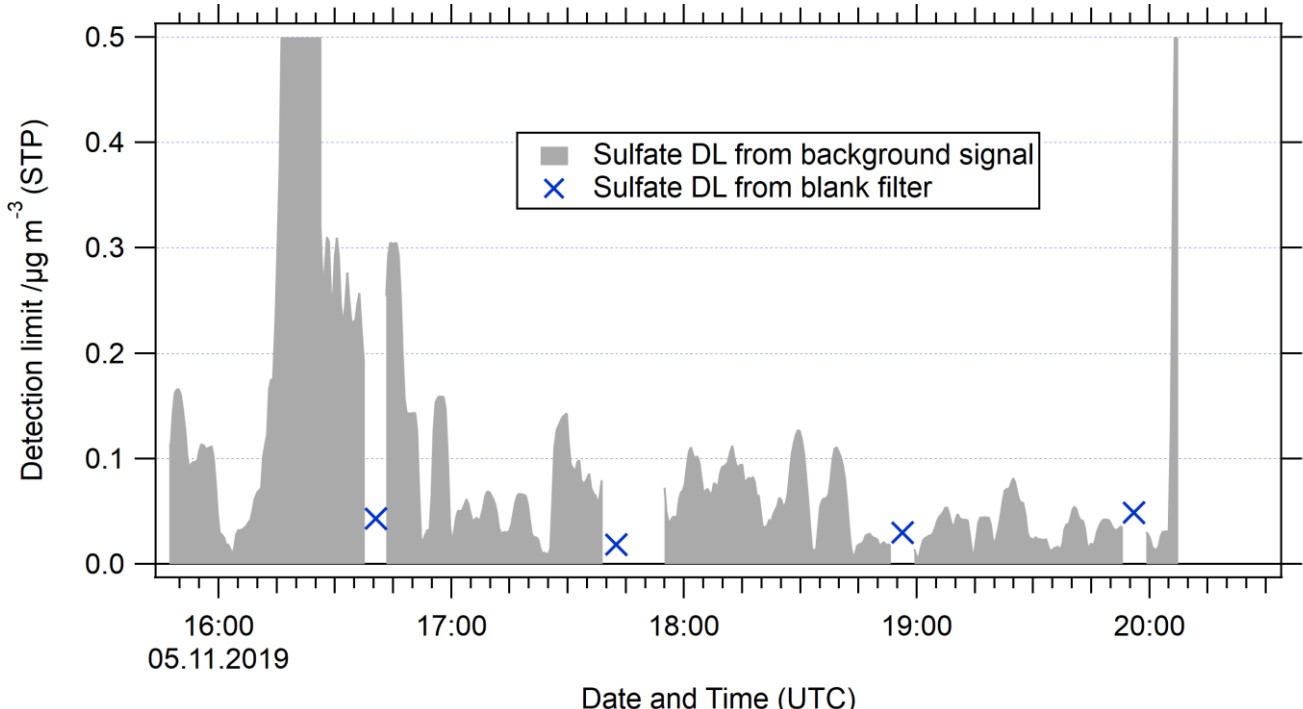

**Figure C1. Comparison between detection limits inferred from the blank filter method (crosses) and the continuous background signal method. At the beginning of the flight, the blank filter method yields a lower DL than the background signal method, while later in the flight, the methods agree better although the blank filter method DL is slightly higher.**

**Data availability**

IAGOS-CARIBIC data are publicly available at ZENODO (Zahn et al., 2024; https://doi.org/10.5281/zenodo.8188548). TPEx data are publicly available at ZENODO (Lachnitt, 2025; http://doi.org/10.5281/zenodo.15371527).

## Author contribution

FR, AL, CS, and JS developed the instrument, CG and SM developed electronic control hard- and software, MH, AW, SB, AZ, HaB, and JS initiated and realized the project, LP, AL, CS, JW, and JS carried out laboratory measurements, PJ, JW, JS, HeB, NE, and PH carried out the TPEx measurements, FO provided $O_3$ data (IAGOS-CARIBIC), TG provided CO data (IAGOS-CARIBIC), HeB, NE, PH provided CO and $O_3$ data (TPEx), JS wrote the manuscript together with CS. All authors contributed to the manuscript.

## Competing interests

Johannes Schneider and Andreas Zahn are members of the editorial board of Atmospheric Measurement Techniques.

## Acknowledgements

This work received funding by the Deutsche Forschungsgemeinschaft (DFG, German Research Foundation), project ID 237520285 and 392638875, and via TRR 301 – Project-ID 428312742.

The original mini-AMS was jointly purchased by internal funds of MPIC and TROPOS.

We would like to thank the following persons and teams who contributed to this work: Frank Helleis, Mark Lamneck, Thomas Böttger and Philipp Schuhmann (MPIC), the IAGOS-CARIBIC teams at KIT and MPIC, Carl Brenninkmeijer (MPIC), the whole IAGOS team (including IAGOS-CORE and IAGOS-CARIBIC), Mike Cubison (Tofwerk), Paul Stock and Helmut Ziereis (DLR-IPA), Harald Franke and Stefan Hofmann (enviscope GmbH), and the GFD and the whole TPEx team.

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
