# Peer review of "CARIBIC-AMS: A fully automated aerosol mass spectrometer for operation on routine passenger flights (IAGOS-CARIBIC): Instrument description and first flight application in the UTLS"

_EGUsphere, 2024_

## Referee Comment (RC2)

**General comments**

This manuscript provides a first characterization of the CARIBIC-AMS, an airborne instrument based on commercial Aerodyne mini-AMS that is part of the IAGOS-CARIBIC research payload. Unlike other airborne particle mass spectrometers that can rely on human input during or at least after each flight, this particular instrument is required to operate in a fully autonomous manner over the course of several flights. Given that IAGOS-CARIBIC operates exclusively on commercial airliners, the reliability of power and communications is typically a lot worse than on most research platforms. While the instrument testing reported here was cut short by COVID, this work does report successful operation for a good portion of that testing period (36 flights, plus a recent campaign in 2024 on a different platform, which is unambiguously a great achievement, for which I want to congratulate the authors

As noted above, this is not a new instrument, but the adaptation of a commercial instrument that has/will operate on a long-term mission to sample the global UT/LS (and some random snippets of the lower troposphere). Hence for a characterization paper like this I would expect it to describe, in addition to the actual modifications, both the operations and analytical performance in enough detail to serve as a guide for future data users/modelers to use the data properly and have a handle on the uncertainties. This paper is fairly detailed on both the motivation and modifications but has a few gaps when describing the operations (the reader learns a great deal on the general CARIBIC schedule of operations, but rather less on the specifics of the CARIBIC-AMS operation). However, it does not describe the analytical performance. While the pre-deployment calibrations are described in detail, no field calibrations are shown. And in lieu of instrumental validation by multiflight instrumental comparisons and performance metrics, the paper chooses to narratively highlight the sulfate measurement on two flight segments. Detection limits are shown and mentioned in the abstract, but it is unclear if these are specific to the single flight segment shown (so "best of") or an actual average of the usable flight segments.

The authors explicitly write in the conclusions that the "metrology paper" will be next. And I certainly agree that it makes sense to leave most of these details for a paper written on the new version of the instrument that is going to start flying shortly on the new CARIBIC platform. Nevertheless, in this manuscript some metrics need to be shown to support that despite all the operational challenges, the instrument actually quantifies PM consistently. Showing, as the authors do in Section 3, examples of AMS sulfate tracking stratospheric markers does not really do that, it just shows that the instrument recorded data and there were no inlet leaks. What needs to be shown is a) how well did the CARIBIC-AMS data agree with collocated instruments and b) are there systematic biases due to the way it is operated (e.g. does the instrument do worse or better on Flight #6 vs Flight #1 of a sequence?). This does not need to be shown for every possible metric (that's paper #2); e.g. just focusing on the (mentioned in the text) agreement between the AMS and the filter sampler would likely be sufficient. But without that, it is impossible to evaluate if the modifications were successful in terms of making the CARIBIC-AMS a quantitative instrument.

I also find some of the AMS specific details to be confusing, and hope that the authors can clarify them, given that this paper is probably going to be read widely.

This paper describes the first deployment of an important new analytical resource to investigate the UT/LS, and therefore is certainly a good fit for AMT.

**Major comments:**

- Characterization of the particle range sampled by the instrument: For any model/measurement or two instrument comparison, a well characterized operational particle range is key. For the CARIBIC-AMS, that would be a product of the aircraft inlet transmission, the transmission of transfer plumbing and of the AMS inlet (both their custom CPI and the PM1 lens). For the aircraft inlet, the authors reference Herrman et al 2001 and Brenninkmeijer 2007, which given the submicron focus of the instrument is perfectly adequate. However, there is no characterization of either plumbing losses or AMS inlet losses. It is mentioned in L128 that the plumbing losses were calculated, but no results are shown. Regarding the AMS inlet losses:
    - For the actual AMS lens, Liu et al, 2007 is referenced. As discussed in e.g. Knote et al, 2011, there are significant discrepancies both at the high and low end of the curve between Liu et al, 2007 and other reports of AMS transmission. Given that the accumulation mode in the stratosphere goes well above the $PM_1$ lens cut (e.g. Brock et al, 2021, Fig. 11), while in the upper troposphere the sampling of particle growth events on the low end of the transmission range is expected, both of these uncertainties will matter for accurate quantification. And they are somewhat dependent on instrument-specific parameters (quality of the alignment, lens vintage, lens pressure), hence an instrument specific characterization is needed, or , in the short term, at least some spot-check confirmation that the actual transmission does indeed follow one of the reference curves.
    - Then there is the issue of the relatively novel CPI used in this work. While Molleker et al 2020 does characterize the transmission performance of the $CPI+PM_{2.5}$ lens in detail, I am not aware of a similar characterization for this $CPI+PM_1$ lens, so the exact impact of the CPI on the lens transmission above is unclear. The authors do report that the comparisons in Mei et al, 2020 suggest serious losses at low altitudes. While I agree that this is unlikely to matter for the portions of the atmosphere sampled by the CARIBIC-AMS, it does (as the authors acknowledge) matter for understanding the sensitivity calibrations on the ground. The authors write: "For this purpose, the mass-based IE calibration as described above is done at various pressures and a pressure-dependent IE is used for the data evaluation" (L287-289). Since said calibration is done at a constant calibrant size, this seems to imply that the authors assume the losses in the CPI at high pressure to be size-independent. This is to me a counterintuitive assumption

that needs more support/explanation, since it's central to the instrument's performance.

- o Again, while these might seem like minor points, if we take Fig 11-l in Brock et al, 2021, there is a ~40% difference in accumulation sulfate mass if one were to apply the Liu vs Knote lens transmission. This uncertainty is not reflected in the 35% accuracy estimate by Bahreini et al, 2009, is dependent on the specific airmass sampled and hence should be treated (and discussed) separately.

  If none of this calibration data is available, then at least the effect of the various literature transmission curves and the "pressure dependent IE" on the final reported concentration needs to be discussed when compared to other instruments.

- Instrumental Collection Efficiency: The current manuscript does not mention collection efficiency (CE) (Canagaratna et al, 2007) at all, which for an AMS-style instrument sampling the free troposphere is a strange omission. This might be warranted if the instrument had a capture vaporizer (Xu et al, 2017, Hu et al, 2017) that mostly obviates the need for a particle bounce correction factor. Now, the authors do not specify what vaporizer was used (please add this!) but both Fig 1 and the mention of a "tungsten vaporizer" strongly implies that this is a standard vaporizer as described in Canagaratna et al 2007. Hence a bounce correction (Middlebrook et al, 2012) is needed for this instrument (and for all the data presented in Section 3) and should be described and discussed. The authors might also want to explain in this context the reasons for that particular vaporizer choice, which seems counterintuitive to me.

  One item I would like the authors to specifically address in this context is how the high ammonium detection limits of the CARIBIC-AMS will affect the accuracy of the CE correction at different altitudes. While averaging can be used to improve the CE accuracy in the stratosphere, this is probably not the case during ascent/descent. If so, what are the consequences for data reporting?

- Sensitivity Calibrations: Section 2.3.2. summarizes some pre-deployment calibrations. Fig 5 and 6a certainly have value in terms of showing that the instrument post reconfiguration was working well. But as noted, for a field instrument it would be much more valuable to show the timeseries (or set of regressions in Fig 5) for the in-field calibrations between 2018 and 2020 to assess instrument stability (particularly the calibration pairs before/after one set of flights). If the testing was too rough on the instrument, this might not be possible, but again, with 36 flights there should be some good data to show.

  The purpose of Fig 6b is not clear. First of all, the uncertainty bands shown seem incorrect. This is a test of RIE fidelity, and the uncertainty for those is 15% (2 sigma) per Bahreini et al, not 35% (which is the total uncertainty). Secondly, previous work has shown no such discrepancy for mixtures (e.g. Jimenez et al, 2016, Xu et al, 2018). Hence the reasons for this disagreement, if real, need to be explained better. There could be issues with different transmission efficiencies for both instruments, different choices in

acquisition cycle, or something else entirely. I think it could also be removed, since it does not really provide much context on the overall instrument performance (a correlation/dual timeseries of ambient measurements on the ground might in any case be more appropiate).

- Size resolved measurements and AMS operation. The description of the ePToF custom implementation and subsequent calibrations is good, although I would suggest to the authors to consider using an alternative figure that shows the (significant) increase in S/N and resolution a bit more convincingly. Also, it should probably be noted somewhere that the theoretical resolution is 1/127 (Hadamard sequence length). However, as the paper is written it is completely unclear if any size dependent measurements were taken in flight. The larger point is that there could be a bit more detail on exactly how the AMS acquisition is run. The authors write that the typical MS acquisition is 30 seconds long, but it is unclear how many open/closed cycles this encompasses and what the effective dutycycle for sampling is. When are the ePToF runs scheduled? And does the instrument operate on a fixed time basis or not?

- Detection limits: The estimation of the detection limits from the closed signal seems very reasonable, but since the instrument takes automated blanks it would be good to show a comparison of the calculated DLs with the ones derived from the statistics of the blanks. More generally, while it is great that the authors reported these DLs, as noted above it is very unclear for what conditions these are specified Do these only apply to Flight 508? I think it would be a lot more informative to:
  a) Report average DLs for the stable stratospheric periods of all available flights
  b) Show the change of these DLs for a range of flights. Ideally other species besides sulfate should be shown, since e.g. ammonium and OA are much more likely to greatly improve with additional pumping over the course of the flight/circuit.

For the abstract, again it would be good to specify what these DLs refer to.It might be simpler and easier to follow to just write that the DLs will scale with $1/\sqrt{N}$ for longer averaging times, instead of quoting 5 min DLs

- Section 3: Validation. Section 3.1 puts the measurements in an appropriate context and helps the reader re-digest the information from the Ops section, but I would suggest adding some AMS context from my previous comments (e.g. discussion of CE and particle size range). Section 3.2 on the other hand does not really introduce anything new. Instead, as noted in the general comments, this would be the place where some instrument comparisons over several flights are shown and discussed (either filters or volume size distributions). Given the teething problems of the instrument, this data is likely going to be noisy and will suggest low accuracies that will hopefully soon be superseded by version 2 of the instrument. This is understandable and can be properly contextualized in the text, but the data should still be shown.

**Minor comments:**

- Single Ion Calibration: This is a minor point, but I am not following the need for periodic single ion calibrations during flight. The single ion calibration at this point (post-2014) just supplies a scaling factor (the SI) to convert signal to counts per second. Historically, this calibration also included the spectral baseline fit, but that has been automated since 2016 and is taken on a per-run basis. Now, the in-flight SI is needed to scale it to the SI of the sensitivity calibration. BUT once the instrument is running and the SI has been characterized once, the variability of the SI is fully captured by the airbeam variability (especially in an instrument with a well-working CPI). So while the ability to do this calibration on the fly e.g. in case of in-flight instrument reboot is important and impressive to have implemented, I would really appreciate it if the authors could explain their rationale for why periodic SI calibrations are needed. As a side note, it would be nice to see a figure with either the airbeam or SI change for a couple of flights.
- Figure 3: It would be very helpful if the different operation states described in the text are added to the figure (maybe as shaded bars). Also, please clarify the units for the Y-axis. Are these A at 24 V? It might be clearer to use W(atts) here.
- Section 2.2.3 is not a straightforward read, since it tries to combine both topological and process details into one narrative. One suggestion to make it more accessible is to add a simple diagram showing the network/software topology/hierarchy and referring to it in the text.
- The units used in the paper are $\mu g\ m^{-3}$ STP. STP is not defined anywhere, and since the atmospheric community is not exactly known for its general adherence to SI units I would strongly urge the authors to spell out what they mean in the text.
- Line 19: Instead of "part" maybe "module" would read better?
- Line 25: "Due to **the** short time"
- Line 86: I would recommend adding the mission description paper for Atom here as well for completeness:
  C. R. Thompson, S. C. Wofsy, M. J. Prather, P. A. Newman, T. F. Hanisco, T. B. Ryerson, D. W. Fahey, E. C. Apel, C. A. Brock, W. H. Brune, K. Froyd, J. M. Katich, J. M. Nicely, J. Peischl, E. Ray, P. R. Veres, S. Wang, H. M. Allen, E. Asher, H. Bian, D. Blake, I. Bourgeois, J. Budney, T. Paul Bui, A. Butler, P. Campuzano-Jost, C. Chang, M. Chin, R. Commane, G. Correa, J. D. Crounse, B. Daube, J. E. Dibb, J. P. Digangi, G. S. Diskin, M. Dollner, J. W. Elkins, A. M. Fiore, C. M. Flynn, H. Guo, S. R. Hall, R. A. Hannun, A. Hills, E. J. Hintsa, A. Hodzic, R. S. Hornbrook, L. Greg Huey, J. L. Jimenez, R. F. Keeling, M. J. Kim, A. Kupc, F. Lacey, L. R. Lait, J.-F. Lamarque, J. Liu, K. Mckain, S. Meinardi, D. O. Miller, S. A. Montzka, F. L. Moore, E. J. Morgan, D. M. Murphy, L. T. Murray, B. A. Nault, J. Andrew Neuman, L. Nguyen, Y. Gonzalez, A. Rollins, K. Rosenlof, M. Sargent, G. Schill, J. P. Schwarz, J. M. St. Clair, S. D. Steenrod, B. B. Stephens, S. E. Strahan, S. A. Strode, C. Sweeney, A. B. Thames, K. Ullmann, N. Wagner, R. Weber, B. Weinzierl, P. O. Wennberg, C. J. Williamson, G. M. Wolfe, L. Zeng, THE NASA ATMOSPHERIC TOMOGRAPHY (ATom) MISSION: Imaging the Chemistry of the Global Atmosphere. *Bull. Am. Meteorol. Soc.* **1**, 1–53 (2021).
- L161: "**into** one new housing"

- L244: "Thus, flow calibrations and size calibration will not change, need to be calibrated only once, and later on only have to be checked.". Agreed. But can you elaborate on how often these checks are done in practice?
- L387: "…than **at** the end of the flight"

**References:**

P. S. K. Liu, R. Deng, K. A. Smith, L. R. Williams, J. T. Jayne, M. R. Canagaratna, K. Moore, T. B. Onasch, D. R. Worsnop, T. Deshler, Transmission efficiency of an aerodynamic focusing lens system: Comparison of model calculations and laboratory measurements for the Aerodyne Aerosol Mass Spectrometer. *Aerosol Sci. Technol.* **41**, 721–733 (2007).

C. Knote, D. Brunner, H. Vogel, J. Allan, A. Asmi, M. Äijälä, S. Carbone, H. D. van der Gon, J. L. Jimenez, A. Kiendler-Scharr, C. Mohr, L. Poulain, A. S. H. Prévôt, E. Swietlicki, B. Vogel, Towards an online-coupled chemistry-climate model: evaluation of trace gases and aerosols in COSMO-ART. *Geoscientific Model Development* **4**, 1077–1102 (2011).

C. A. Brock, K. D. Froyd, M. Dollner, C. J. Williamson, G. Schill, D. M. Murphy, N. J. Wagner, A. Kupc, J. L. Jimenez, P. Campuzano-Jost, B. A. Nault, J. C. Schroder, D. A. Day, D. J. Price, B. Weinzierl, J. P. Schwarz, J. M. Katich, S. Wang, L. Zeng, R. Weber, J. Dibb, E. Scheuer, G. S. Diskin, J. P. DiGangi, T. Bui, J. M. Dean-Day, C. R. Thompson, J. Peischl, T. B. Ryerson, I. Bourgeois, B. C. Daube, R. Commane, S. C. Wofsy, Ambient aerosol properties in the remote atmosphere from global-scale in situ measurements. *Atmos. Chem. Phys.* **21**, 15023–15063 (2021).

R. Bahreini, B. Ervens, A. M. Middlebrook, C. Warneke, J. A. de Gouw, P. F. DeCarlo, J. L. Jimenez, C. A. Brock, J. A. Neuman, T. B. Ryerson, H. Stark, Atlas, E, J. Brioude, A. Fried, J. S. Holloway, J. Peischl, D. Richter, J. Walega, P. Weibring, A. G. Wollny, F. C. Fehsenfeld, Organic aerosol formation in urban and industrial plumes near Houston and Dallas, Texas. *J. Geophys. Res.* **114**, D00F16-D00F16 (2009).

S. Molleker, F. Helleis, T. Klimach, O. Appel, H.-C. Clemen, A. Dragoneas, C. Gurk, A. Hünig, F. Köllner, F. Rubach, C. Schulz, J. Schneider, S. Borrmann, Application of an O-ring pinch device as a constant-pressure inlet (CPI) for airborne sampling. *Atmospheric Measurement Techniques* **13**, 3651–3660 (2020).

F. Mei, J. Wang, J. M. Comstock, R. Weigel, M. Krämer, C. Mahnke, J. E. Shilling, J. Schneider, C. Schulz, C. N. Long, M. Wendisch, L. A. T. MacHado, B. Schmid, T. Krisna, M. Pekour, J. Hubbe, A. Giez, B. Weinzierl, M. Zoeger, M. L. Pöhlker, H. Schlager, M. A. Cecchini, M. O. Andreae, S. T. Martin, S. S. De Sá, J. Fan, J. Tomlinson, S. Springston, U. Pöschl, P. Artaxo, C. Pöhlker, T. Klimach, A. Minikin, A. Afchine, S. Borrmann, Comparison of aircraft measurements during GoAmazon2014/5 and ACRIDICON-CHUVA. *Atmospheric Measurement Techniques* **13**, 661–684 (2020).

M. R. Canagaratna, J. T. Jayne, J. L. Jimenez, J. D. Allan, M. R. Alfarra, Q. Zhang, T. B. Onasch, F. Drewnick, H. Coe, A. Middlebrook, A. Delia, L. R. Williams, A. M. Trimborn, M. J. Northway, P. F. DeCarlo, C. E. Kolb, P. Davidovits, D. R. Worsnop, Chemical and microphysical characterization of ambient aerosols with the aerodyne aerosol mass spectrometer. *Mass Spectrom. Rev.* **26**, 185–222 (2007).

A. M. Middlebrook, R. Bahreini, J. L. Jimenez, M. R. Canagaratna, Evaluation of Composition-Dependent Collection Efficiencies for the Aerodyne Aerosol Mass Spectrometer using Field Data. *Aerosol Sci. Technol.* **46**, 258–271 (2012).

J. L. Jimenez, M. R. Canagaratna, F. Drewnick, J. D. Allan, M. R. Alfarra, A. M. Middlebrook, J. G. Slowik, Q. Zhang, H. Coe, J. T. Jayne, D. R. Worsnop, Comment on "The effects of molecular weight and thermal decomposition on the sensitivity of a thermal desorption aerosol mass spectrometer." *Aerosol Sci. Technol.* **50**, i–xv (2016).

W. Xu, A. Lambe, P. Silva, W. Hu, T. Onasch, L. Williams, P. Croteau, X. Zhang, L. Renbaum-Wolff, E. Fortner, J. L. Jimenez, J. Jayne, D. Worsnop, M. Canagaratna, Laboratory evaluation of species-dependent relative ionization efficiencies in the Aerodyne Aerosol Mass Spectrometer. *Aerosol Sci. Technol.* **52**, 626–641 (2018).

---

## Author Response (AR1)

**egusphere-2024-3969**

Schneider et al. "CARIBIC-AMS: A fully automated aerosol mass spectrometer for operation on routine passenger flights (IAGOS-CARIBIC): Instrument description and first flight application in the UTLS"

**Reply to Reviewer #1**

Reviewer comments in black

Our replies in red
Changes in manuscript in blue

This manuscript describes the modification of a commercial aerosol mass spectrometer for fully autonomous operation on board a commercial jetliner and provides some example data resulting from some flights. The scientific goals and context are well-motivated. The manuscript is clearly organized and well-written. I have a few minor comments I would like the authors to address, but otherwise I find this suitable to publish.

We thank the reviewer for this positive rating of our manuscript

Given the description of the use of the new particle time-of-flight mode, I would've expected to see some particle size-dependent example data in this manuscript, or an explanation of why none was included. Was this mode used in flight? Is there any promise for new, useful results from the size-resolved data?

We have not used the ePToF mode in flight yet. The synchronization between the ToF pulser and the chopper was not ready during the IAGOS-CARIBIC flights between 2018 and 2020. For the TPEx mission, we did not use ePToF but decided to maximize the time spent in mass spectrum mode to obtain the best possible signal-to-noise ratio with a highest possible time resolution. The reason for this was that the objective of TPEx was small scale mixing at the tropopause.

We are confident that the ePToF mode will be operating reliably in future application. When IAGOS-CARIBIC continues the airborne operation, we expect important results from size-resolved data.

We added this information to the revised version:

The size distribution mode has not been used during aircraft operation yet. During the IAGOS-CARIBIC flight between 2018 and 2020, the development of the electronic synchronization had not been completed. For TPEx, it was decided to focus on mass concentrations only to obtain a higher signal-to-noise ratio together with the best possible time resolution. This was required to meet the objectives of the campaign which focused on small-scale mixing processes in the tropopause region.

Line 346-354. (Why) should we expect the ePTOF mode to result in a narrower distribution of particle transit times? Is it just improved statistics for a distribution that is fundamentally the same? Do you get the same central time of flight estimate for both modes?

The graph in Figure 7 shows the raw data as they are measured. We assume that the narrower distribution is a result of better statistics.

From the Gauss fit to the peaks we obtain the same central time of flight: PToF: 2545 µs, ePToF: 2555 µs. This difference of 10 µs is in the order of the resolution of the measurement (12 µs)

We added the information on the central time of flight to the manuscript:

The particles arrive after 2400 µs at the vaporizer and the maximum of the signal is at about 2500 µs (2545 µs for PToF and 2555 µs for ePToF).

Section 3.1.1 it would be useful to know the detection limits of the instrument under optimal conditions in the lab as well.

In the laboratory, we measured (August 2024) the following detection limits:

$NO_3$: 0.056 µg m$^{-3}$ STP
$SO_4$: 0.025 µg m$^{-3}$ STP
Organics: 0.25 µg m$^{-3}$ STP
$NH_4$: 0.35 µg m$^{-3}$ STP
Chloride: 0.019 µg m$^{-3}$ STP

This applies to the following settings:

Pressure in front of the CPI: 300 hPa
Ionization efficiency: $6.8 \times 10^{-8}$ ions/molecule
Time step: 10 seconds (total run time: 25 seconds, thereof 15 in ePToF mode and 10 in MS mode)

These values, converted to 30 sec, are comparable to the in-flight values for nitrate, chloride and ammonium, but better for sulfate and organics.

Also with respect to questions by reviewer #2, we moved the detection limit discussion into a new subchapter and expanded Table 1, including also laboratory DLs given above:

In the laboratory, lower DL values are achieved. Data measured in August 2024 with 10 seconds time resolution are also included in Table 1. These values are comparable to the in-flight values for nitrate, chloride and ammonium, but better for sulfate and organics. During the TPEx campaign, detection limits were higher, likely due to the short pumping time before the flights and the short flight duration of about 3-4 hours. Only on flight days where two flights after each other were conducted, the conditions for reaching good vacuum conditions were slightly better. The calculated detection limits for this flight (F08 on 17 June 2024) are included in Table 1.

|  | In-flight IAGOS-CARIBIC | | | In-flight TPEx F08 | Laboratory | |
|  | 30 sec | 5 min | 30 min | 30 sec | 10 sec | 30 sec |
|---|---|---|---|---|---|---|
| Organics | 0.69 | 0.22 | 0.089 | 1.56 | 0.25 | 0.14 |
| Nitrate | 0.055 | 0.017 | 0.007 | 0.16 | 0.056 | 0.032 |
| Sulfate | 0.035 | 0.011 | 0.005 | 0.10 | 0.025 | 0.014 |
| Ammonium | 0.38 | 0.12 | 0.049 | 1.31 | 0.35 | 0.20 |
| Chloride | 0.022 | 0.007 | 0.003 | 0.10 | 0.019 | 0.011 |

**Reply to Reviewer #2**

Reviewer comments in black

Our replies in red
Changes in manuscript in blue

**General comments**
This manuscript provides a first characterization of the CARIBIC-AMS, an airborne instrument based on commercial Aerodyne mini-AMS that is part of the IAGOS-CARIBIC research payload. Unlike other airborne particle mass spectrometers that can rely on human input during or at least after each flight, this particular instrument is required to operate in a fully autonomous manner over the course of several flights. Given that IAGOS-CARIBIC operates exclusively on commercial airliners, the reliability of power and communications is typically a lot worse than on most research platforms. While the instrument testing reported here was cut short by COVID, this work does report successful operation for a good portion of that testing period (36 flights, plus a recent campaign in 2024 on a different platform, which is unambiguously a great achievement, for which I want to congratulate the authors

As noted above, this is not a new instrument, but the adaptation of a commercial instrument that has/will operate on a long-term mission to sample the global UT/LS (and some random snippets of the lower troposphere). Hence for a characterization paper like this I would expect it to describe, in addition to the actual modifications, both the operations and analytical performance in enough detail to serve as a guide for future data users/modelers to use the data properly and have a handle on the uncertainties. This paper is fairly detailed on both the motivation and modifications but has a few gaps when describing the operations (the reader learns a great deal on the general CARIBIC schedule of operations, but rather less on the specifics of the CARIBICAMS operation). However, it does not describe the analytical performance. While the predeployment calibrations are described in detail, no field calibrations are shown. And in lieu of instrumental validation by multiflight instrumental comparisons and performance metrics, the paper chooses to narratively highlight the sulfate measurement on two flight segments. Detection limits are shown and mentioned in the abstract, but it is unclear if these are specific to the single flight segment shown (so "best of") or an actual average of the usable flight segments.

The authors explicitly write in the conclusions that the "metrology paper" will be next. And I certainly agree that it makes sense to leave most of these details for a paper written on the new version of the instrument that is going to start flying shortly on the new CARIBIC

platform. Nevertheless, in this manuscript some metrics need to be shown to support that despite all the operational challenges, the instrument actually quantifies PM consistently. Showing, as the authors do in Section 3, examples of AMS sulfate tracking stratospheric markers does not really do that, it just shows that the instrument recorded data and there were no inlet leaks. What needs to be shown is a) how well did the CARIBIC-AMS data agree with collocated instruments and b) are there systematic biases due to the way it is operated (e.g. does the instrument do worse or better on Flight #6 vs Flight #1 of a sequence?). This does not need to be shown for every possible metric (that's paper #2); e.g. just focusing on the (mentioned in the text) agreement between the AMS and the filter sampler would likely be sufficient. But without that, it is impossible to evaluate if the modifications were successful in terms of making the CARIBICAMS a quantitative instrument.

I also find some of the AMS specific details to be confusing, and hope that the authors can clarify them, given that this paper is probably going to be read widely.
This paper describes the first deployment of an important new analytical resource to investigate the UT/LS, and therefore is certainly a good fit for AMT.

Thank you for the detailed review and the suggestion for improvement of the manuscript. The major and minor comments are answered below. Here we respond to the additional questions and remarks in the text above:

To the points a) and b) from above:

a) Agreement with collocated instruments: We include below a comparison with a UHSAS during TPEx for the flight that we presented here. Additionally, we refer to a published preprint presenting the comparison for two other flights during TPEx (Joppe et al., 2025). For the IAGOS-CARIBIC flights, no collocated sizing instruments were available during the period between October 2018 and March 2020. Data from an optical particle counter (OPC) operated on IAGOS-CARIBIC are not available up to now, but would only be of minor use, because the lower size cut of the OPC is 250 nm. The impactor sampler operated by Lund University was not in operation anymore in 2018 and later, and the OPSS (lower size cut 140 nm) was broken.

b) The instrument generally operates better during the later flights of a sequence, since power down time between two flights is generally much shorter that power down time before the first flight (due to transport from our lab to KIT and then to the airport).

To the remark concerning field calibrations: We do not / did not intend to perform field calibrations. The way the operation of IAGOS-CARIBIC is (and has been) planned is the following: The instrument is calibrated in the lab at MPIC. One day before a flight sequence, it is brought to KIT where it is installed into the measurement container which is then transported to the airport and installed into the aircraft. After the flight sequence, the container is transported back to KIT, the AMS is brought back to MPIC where the next calibration is/will be done.

To the remark of "agreement between the AMS and the filter sampler": We assume this refers to lines 401-402 where we stated that the presented example data from November 2019 data "lie in a similar range as previous IAGOS-CARIBIC data"

published between 2013 and 2015. Thus, this is not a direct comparison that we can focus on.

**Major comments:**
• Characterization of the particle range sampled by the instrument: For any model/measurement or two instrument comparison, a well characterized operational particle range is key. For the CARIBIC-AMS, that would be a product of the aircraft inlet transmission, the transmission of transfer plumbing and of the AMS inlet (both their custom CPI and the PM1 lens). For the aircraft inlet, the authors reference Herrman et al 2001 and Brenninkmeijer 2007, which given the submicron focus of the instrument is perfectly adequate. However, there is no characterization of either plumbing losses or AMS inlet losses. It is mentioned in L128 that the plumbing losses were calculated, but no results are shown.

We added a section (2.3.1) describing the transmission efficiency of the aircraft inlet and the sampling lines to the CARBIC-AMS:

**2.3.1. Inlet transmission of IAGOS-CARIBIC operation setup**

The inlet used on the Airbus A340-600 is described in detail in Brenninkmeijer et al. (2007). The aerosol sampling lines inside the container were slightly modified when the CARIBIC-AMS was first integrated in 2017. The inlet sampling efficiency (Fig. 04 a) is estimated based on empirical equations from the literature (Baron and Willeke, 2001) and wind tunnel experiments with another aircraft-borne aerosol inlet (Hermann et al., 2001). The further transport efficiency through the sampling lines from the inlet to the container-laboratory (Fig. 04 b) and to the CARIBIC-AMS CPI (Fig. 04 c) were calculated using the particle loss calculator (von der Weiden et al., 2009), for spherical particles with a density of 1.5 g cm-3. Fig. 04 d) shows the total sampling line transmission from the outside air to the CPI of the CARIBIC-AMS.

The transmission is above 80% in a size range between 40 and 700 nm, corresponding well to the size range of the AMS inlet (about 50 – 800 nm). However, stratospheric aerosol particles may be larger (Brock et al., 2021), such that the upper cut-off of the inlet plays a role for the exact quantification of aerosol mass. For the new IAGOS-CARIBIC setup on the A350, a new inlet design has been developed which will have higher transmission properties for large particles. The influence of the transmission of the CPI together with the aerodynamic lens is discussed in the following section (2.3.2).

[Figure]

**Figure 4. Aerosol inlet transmission efficiency, derived by a combination of measurements and calculations: Transmission of the aircraft inlet (a), the sampling lines to the container-laboratory (b), the sampling lines in the CARIBIC-AMS rack (c), and total transport efficiency (d).**

Regarding the AMS inlet losses:

o  For the actual AMS lens, Liu et al, 2007 is referenced. As discussed in e.g. Knote et al, 2011, there are significant discrepancies both at the high and low end of the curve between Liu et al, 2007 and other reports of AMS transmission. Given that the accumulation mode in the stratosphere goes well above the PM1 lens cut (e.g. Brock et al, 2021, Fig. 11), while in the upper troposphere the sampling of particle growth events on the low end of the transmission range is expected, both of these uncertainties will matter for accurate quantification. And they are somewhat dependent on instrument-specific parameters (quality of the alignment, lens vintage, lens pressure), hence an instrument specific characterization is needed, or , in the short term, at least some spot-check confirmation that the actual transmission does indeed follow one of the reference curves.

We have not performed a measurement of the transmission of the PM1 lens of the CARIBIC-AMS. We had assumed that the transmission measurements of the PM1 lens as published by Liu et al. (2007) are valid also for our lens. We will do this measurement, but this will not be possible in the time frame needed for the revision of this manuscript. See also reply to the following comment.

o  Then there is the issue of the relatively novel CPI used in this work. While Molleker et al 2020 does characterize the transmission performance of the CPI+PM2.5 lens in detail, I am not aware of a similar characterization for this CPI+PM1 lens, so the exact impact of the CPI on the lens transmission above is unclear. The authors do report that the comparisons in Mei et al, 2020 suggest serious losses at low altitudes. While I agree that this is unlikely to matter for the portions of the atmosphere sampled by the CARIBIC-AMS, it does (as the authors acknowledge) matter for understanding the sensitivity calibrations on the ground. The authors write: "For this purpose, the mass-based IE calibration as described above is done at various pressures and a pressure-dependent IE is used for the data evaluation" (L287-289). Since said

calibration is done at a constant calibrant size, this seems to imply that the authors assume the losses in the CPI at high pressure to be size-independent. This is to me a counterintuitive assumption that needs more support/explanation, since it's central to the instrument's performance.

As said before, we have not done the measurement of the inlet (CPI + PM1 lens) transmission, because we relied on published data on the PM1 lens (e.g., Liu et al., 2007) and the CPI with PM2.5 lens (Molleker et al., 2020). However, we agree that the combination of CPI and PM1 lens appears to have a different transmission characteristic than the CPI in combination with the PM2.5 lens. The measurement will be made in our lab as soon as possible, but currently the CARIBIC-AMS needs to undergo some hardware modification for the installation in the new container setup on the A350. Therefore, these measurements need more time and can't be part of the current paper. The comparison with a UHSAS shown below and added to the manuscript indicates that the losses are not severe for altitudes above 8000 m.

We use the same combination of CPI and PM1 lens in the C-ToF-AMS that is used on other aircraft, mainly on HALO (e.g., Schulz et al., 2018; Mei et al., 2020; Reifenberg et al., 2022), and as mentioned above, we find good agreement with other instruments above about 2000 m altitude. However, if it turns out that the combination CPI + PM2.5 lens indeed gives better transmission values, we will consider changing to the PM2.5 lens.

We added the following sentence to section 2.3.2:
However, for vertical profiles this effect needs to be corrected. For this purpose, the mass-based IE calibration as described above is done at various pressures and a pressure-dependent IE is used for the data evaluation. This procedure assumes a size-independent transmission function of the CPI-ADL combination which represents a simplification. A size-resolved transmission measurement of the current CPI-ADL combination has not been done so far, but we assume that it is similar to the results reported by Molleker et al (2020). In-flight comparison with collocated measurements has not been possible during the IAGOS-CARIBIC flights yet, but comparisons during the TPEx campaign in 2024 with volume concentrations inferred from UHSAS data (see section 3.3 and Joppe et al., 2025) show satisfying agreement between both instruments for altitudes above about 8000 m.

o  Again, while these might seem like minor points, if we take Fig 11-l in Brock et al, 2021, there is a ~40% difference in accumulation sulfate mass if one were to apply the Liu vs Knote lens transmission. This uncertainty is not reflected in the 35% accuracy estimate by Bahreini et al, 2009, is dependent on the specific airmass sampled and hence should be treated (and discussed) separately. If none of this calibration data is available, then at least the effect of the various literature transmission curves and the "pressure dependent IE" on the final reported concentration needs to be discussed when compared to other instruments.

It should be noted that Liu et al. (2007) report on laboratory measurements, while Knote et al. (2011) summarizes available transmission measurements and use a linear parameterization of these data. Thus, the "Knote lens transmission" is not necessarily a better or more representative transmission curve than that inferred from individual measurements.
However, we agree that the size distributions in Fig 11-I of Brock et al. (2021) suggest that the upper size cut of the inlet system of the CARIBIC-AMS leads to additional

uncertainty due to inlet transmission, especially for particles in the micrometer size rang as observed in the stratosphere by Brock et al. (2021). On the other hand, many of the IAGOS-CARIBIC data will be recorded in the upper troposphere due to the typical flight altitudes of passenger aircraft where particles are likely smaller than in the lowermost stratosphere.

We included a discussion on the uncertainty by the lens cut-off and the extension of accumulation mode beyond the lens transmission in the new section 2.3.1 (see above):

The transmission is above 80% in a size range between 40 and 700 nm, corresponding well to the size range of the AMS inlet (about 50 – 800 nm). However, stratospheric aerosol particles may be larger (Brock et al., 2021), such that the upper cut-off of the inlet plays a role for the exact quantification of aerosol mass. For the new IAGOS-CARIBIC setup on the A350, a new inlet design has been developed which will have higher transmission properties for large particles. The influence of the transmission of the CPI together with the aerodynamic lens is discussed in the following section (2.3.2).

• Instrumental Collection Efficiency: The current manuscript does not mention collection efficiency (CE) (Canagaratna et al, 2007) at all, which for an AMS-style instrument sampling the free troposphere is a strange omission. This might be warranted if the instrument had a capture vaporizer (Xu et al, 2017, Hu et al, 2017) that mostly obviates the need for a particle bounce correction factor. Now, the authors do not specify what vaporizer was used (please add this!) but both Fig 1 and the mention of a "tungsten vaporizer" strongly implies that this is a standard vaporizer as described in Canagaratna et al 2007. Hence a bounce correction (Middlebrook et al, 2012) is needed for this instrument (and for all the data presented in Section 3) and should be described and discussed. The authors might also want to explain in this context the reasons for that particular vaporizer choice, which seems counterintuitive to me. One item I would like the authors to specifically address in this context is how the high ammonium detection limits of the CARIBIC-AMS will affect the accuracy of the CE correction at different altitudes. While averaging can be used to improve the CE accuracy in the stratosphere, this is probably not the case during ascent/descent. If so, what are the consequences for data reporting?

Yes, we use a standard vaporizer in the CARIBIC-AMS. We added this information in the revised version:

The particle beam is directed onto a standard vaporizer (e. g., Hu et al., 2017) made out of tungsten, operated at 600°C

The reason for choosing a standard vaporizer is the measurement of size distributions. As reported in Hu et al. (2017) and Hu et al. (2020), the capture vaporizer broadens the size distributions. The authors argue that this problem may only be of limited importance for field measurements, but we decided that with including the ePToF mode and the synchronization with the ToF pulser, the best possible chopper configuration for size distribution measurements should be preferred.
However, we don't rule out that we will test a capture vaporizer in future experiments.

The collection efficiency (CE) is applied only during data evaluation and can therefore be adjusted to the hardware setup (e.g. vaporizer type) and also to possible newer

parameterizations. Thus, the CE is not part of the hardware modifications and software procedures required for automation, which are the focus of this paper.
The data examples shown in the paper were calculated with the simple approach of a constant CE of 0.5. We added this information in the revised version:

Section 3.1:
For the data evaluation, we used a constant collection efficiency (CE) of 0.5.

Section 3.2:
Again, a constant collection efficiency of 0.5 was used for the calculation of the aerosol mass loading.

However, for future routine evaluation of IAGOS-CARIBIC data, we will apply the composition dependent CE of Middlebrook et al. (2012). The high detection limit especially of ammonium and the therefore limited time resolution are certainly a problem for the ascent and descent phases.
We included a discussion on this in the revised version in section 3.1:

For the data evaluation, we used a constant collection efficiency (CE) of 0.5. This is a first-guess simplification instead of using the composition-dependent CE (CDCE) recommended by Middlebrook et al. (2012). However, the high detection limits for ammonium impose a problem for determining the correct CDCE, especially under acidic conditions as in the stratosphere, because for this a good ammonium measurement is required. Thus, we likely overestimate sulfate in the stratosphere. On the other hand, the limited particle transmission of the inlet system for particles larger than 800 nm (see Fig. 4) likely leads to an underestimation of stratospheric sulfate, because, as shown in Brock et al. (2021), stratospheric particle size distributions of sulfate may extend to 1 µm and above.

• Sensitivity Calibrations: Section 2.3.2. summarizes some pre-deployment calibrations. Fig 5 and 6a certainly have value in terms of showing that the instrument post reconfiguration was working well. But as noted, for a field instrument it would be much more valuable to show the timeseries (or set of regressions in Fig 5) for the in-field calibrations between 2018 and 2020 to assess instrument stability (particularly the calibration pairs before/after one set of flights). If the testing was too rough on the instrument, this might not be possible, but again, with 36 flights there should be some good data to show.

Here we have to emphasize that we do not perform in-field calibrations. The IAGOS-CARIBIC container is removed from the aircraft after each flight sequence an brought back to the KIT lab. From there, we transported the instrument to the MPIC lab where all calibrations and performance checks are done. Thus, the pre- (or post-) deployment calibrations shown in section 2.3.2 (now 2.3.3) are representative for the standard operation of the instrument.

The purpose of Fig 6b is not clear. First of all, the uncertainty bands shown seem incorrect. This is a test of RIE fidelity, and the uncertainty for those is 15% (2 sigma) per Bahreini et al, not 35% (which is the total uncertainty). Secondly, previous work has shown no such discrepancy for mixtures (e.g. Jimenez et al, 2016, Xu et al, 2018). Hence the reasons for this disagreement, if real, need to be explained better. There could be issues with different transmission efficiencies for both instruments, different choices in acquisition cycle, or something else entirely. I think it could also be removed, since it does not really provide much

context on the overall instrument performance (a correlation/dual timeseries of ambient measurements on the ground might in any case be more appropiate).

Thanks for pointing out the mistake in the uncertainty for the RIE. Bahreini et al. (2009) in fact report 10% for ammonium and 15% for sulfate. However, the comparison for nitrate also needs an uncertainty, which following this argumentation would not exist, by definition of RIE.

We therefore included the overall 35% uncertainty for the nitrate comparison and a 15% uncertainty range for the RIE comparison in Figure 6b:

[Figure]

We still think that the figure is useful because it shows the reproducibility of two different AMS instruments. To our opinion, a comparison with well defined (350 nm) laboratory generated aerosol particles appears favourable over a comparison with unknown ambient aerosol.

• Size resolved measurements and AMS operation. The description of the ePToF custom implementation and subsequent calibrations is good, although I would suggest to the authors to consider using an alternative figure that shows the (significant) increase in S/N and resolution a bit more convincingly. Also, it should probably be noted somewhere that the theoretical resolution is 1/127 (Hadamard sequence length). However, as the paper is written it is completely unclear if any size dependent measurements were taken in flight.
The larger point is that there could be a bit more detail on exactly how the AMS acquisition is run. The authors write that the typical MS acquisition is 30 seconds long, but it is unclear how many open/closed cycles this encompasses and what the effective dutycycle for sampling is. When are the ePToF runs scheduled? And does the instrument operate on a fixed time basis or not?

The intention of Figure 7 was to show the raw data as they were measured for conditions representative for the upper troposphere (p = 300 hPa, particle concentration 100 cm$^{-3}$). That the increase in S/N for a short sampling time of 8

seconds is not as significant as might have been expected is surprising but cannot be ignored here.

As mentioned above in the reply to reviewer #1, we have not used the ePToF mode in flight yet. The synchronization between the ToF pulser and the chopper was not ready during the IAGOS-CARIBIC flights between 2018 and 2020. For the TPEx mission, we did not use ePToF but decided to maximize the time spent in mass spectrum mode to obtain the best possible signal-to-noise ratio with a highest possible time resolution. The reason for this was that the objective of TPEx was small scale mixing at the tropopause.

We have clarified this in the revised version:

The size distribution mode has not been used during aircraft operation yet. During the IAGOS-CARIBIC flights between 2018 and 2020, the development of the electronic synchronization had not been completed. For TPEx, it was decided to focus on mass concentrations only to obtain a higher signal-to-noise ratio together with the best possible time resolution. This was required to meet the objectives of the campaign which focused on small-scale mixing processes in the tropopause region.

Regarding the timing:

For flights when we tried ePToF, we used 20 seconds in MS mode and 10 seconds in ePToF mode. In the MS mode we used 5 seconds "open" and 5 seconds "closed".

In flights without trying ePToF (because we had realized before that it won't work) we used 20 seconds time resolution, i.e. also twice 5 sec open and 5 secs closed.

The sampling dutycycles were 72% for the 30 sec saving with ePToF and 75% for the 20 sec saving w/o ePToF.

During the IAGOS-CARIBIC flights from 2018 to 2020, the instrument did not operate on a fixed time base.

However, it must be emphasized that these settings are subject to change based on improvements in detection limit and sensitivity, as well as from experience and results from future flight application. The IAGOS database can cope with different time resolution settings, so there is no need to define and fix the time resolution and time steps in advance. Therefore, we don't give this information here in the instrument paper but will include it in future publications of individual data sets.

• Detection limits: The estimation of the detection limits from the closed signal seems very reasonable, but since the instrument takes automated blanks it would be good to show a comparison of the calculated DLs with the ones derived from the statistics of the blanks. More generally, while it is great that the authors reported these DLs, as noted above it is very unclear for what conditions these are specified Do these only apply to Flight 508? I think it would be a lot more informative to:
a) Report average DLs for the stable stratospheric periods of all available flights

b) Show the change of these DLs for a range of flights. Ideally other species besides sulfate should be shown, since e.g. ammonium and OA are much more likely to greatly improve with additional pumping over the course of the flight/circuit.

For the abstract, again it would be good to specify what these DLs refer to. It might be simpler and easier to follow to just write that the DLs will scale with 1/sqrt(N) for longer averaging times, instead of quoting 5 min DLs

It has been explained in Drewnick et al. (2009) that the continuous method for determining AMS detection limits by using the background signal yields more realistic results than the filter method. The main purpose of the blank filter is to determine the gas-phase contributions to certain ions.

In the first flight on November 5, 2019 (Fig. 9 in the preprint), the instrument took 4 blank filters. The detection limits calculated (after Drewnick et al., 2009) as $3 \times \sigma_{Filter}$ are similar but lower than those inferred from the background signal mode:

[Figure]

The reason for the suggested restriction to stratospheric periods for the detection limits is not clear to us. To our opinion, the detection limits depend on the background in the vacuum chamber and by that mainly on pumping time. The DLs should not depend on altitude or ambient pressure (because the CPI keeps the inlet mass flow constant), and also not on the composition of the ambient air. Maybe the higher ozone volume mixing ratio in the stratosphere plays a role, but that is unlikely and speculative. An analysis of stratospheric flight parts versus upper tropospheric flight is not subject of this technical paper.

We modified the abstract according the reviewer's suggestion and report only the 30-sec DL.

Due to the short time for evacuation of the vacuum chamber to sufficiently low pressures before measurement, detection limits are higher during regular flights than during ground operation and were determined to be 0.035 µg m$^{-3}$ STP (sulfate), 0.055 µg m$^{-3}$ STP (nitrate), 0.69 µg m$^{-3}$ STP (organics), 0.38 µg m$^{-3}$ STP (ammonium) and 0.022 µg m$^{-3}$ STP (chloride) for a time resolution of 30 seconds. Since the IAGOS-CARIBIC project aims for climatological, regular, long-term data, longer data averaging times are possible, thereby lowering the detection limits by the square root of the number of averaged data points.

We moved the detection limit discussion into a new subsection (3.3) and included more values in Table 1:

In the laboratory, lower DL values are achieved. Data measured in August 2024 with 10 seconds time resolution are also included in Table 1. These values are comparable to the in-flight values for nitrate, chloride and ammonium, but better for sulfate and organics. During the TPEx campaign, detection limits were higher, likely due to the short pumping time before the flights and the short flight duration of about 3-4 hours. Only on flight days where two flights after each other were conducted, the conditions for reaching good vacuum conditions were slightly better. The calculated detection limits for this flight (F08 on 17 June 2024) are included in Table 1.

| | In-flight IAGOS-CARIBIC | | | In-flight TPEx F08 | Laboratory | |
|---|---|---|---|---|---|---|
| | 30 sec | 5 min | 30 min | 30 sec | 10 sec | 30 sec |
| Organics | 0.69 | 0.22 | 0.089 | 1.56 | 0.25 | 0.14 |
| Nitrate | 0.055 | 0.017 | 0.007 | 0.16 | 0.056 | 0.032 |
| Sulfate | 0.035 | 0.011 | 0.005 | 0.10 | 0.025 | 0.014 |
| Ammonium | 0.38 | 0.12 | 0.049 | 1.31 | 0.35 | 0.20 |
| Chloride | 0.022 | 0.007 | 0.003 | 0.10 | 0.019 | 0.011 |

- Section 3: Validation. Section 3.1 puts the measurements in an appropriate context and helps the reader re-digest the information from the Ops section, but I would suggest adding some AMS context from my previous comments (e.g. discussion of CE and particle size range). Section 3.2 on the other hand does not really introduce anything new.
Instead, as noted in the general comments, this would be the place where some instrument comparisons over several flights are shown and discussed (either filters or volume size distributions). Given the teething problems of the instrument, this data is likely going to be noisy and will suggest low accuracies that will hopefully soon be superseded by version 2 of the instrument. This is understandable and can be properly contextualized in the text, but the data should still be shown.

We added a discussion of CE and size range to section 3.1 in the revised version:

For the data evaluation, we used a constant collection efficiency (CE) of 0.5. This is a first-guess simplification instead of using the composition-dependent CE (CDCE) recommended by Middlebrook et al. (2012). However, the high detection limits for ammonium impose a problem for determining the correct CDCE, especially under acidic conditions as in the stratosphere, because for this a good ammonium measurement is required. Thus, we likely overestimate sulfate in the stratosphere. On the other hand, the limited particle transmission of the inlet system for particles larger than 800 nm (see Fig. 4) likely leads to an underestimation of stratospheric sulfate, because, as shown in Brock et al. (2021), stratospheric particle size distributions of sulfate may extend to 1 µm and above.

Section 3.2 was indented to demonstrate the operation of the instrument under research campaign conditions in a manual operation mode, mainly because between March 2020 and summer 2024 there was no other opportunity to operate the instrument on an aircraft. Furthermore, with respect to validation and comparison, unfortunately during the IAGOS-CARIBIC flights between Oct 2018 to March 2020, the OPSS (Hermann et al., 2016) of TROPOS was not operational.

The optical particle counter operated by FZ Julich detects particles only above a diameter of 250 nm. This may still be useful for comparison, but the data are not publically available in full detail on the Zenodo data base (Zahn et al., 2025) and are still not available.

During the TPEx mission, a UHSAS was operated in parallel to the CARIBIC-AMS. We converted the AMS mass concentration data with the density inferred from the composition to a volume concentration. The AMS data were evaluated with a constant CE of 0.5. The comparison is now included in Figure 11 (formerly 10). Also, a reference to Joppe et al. (2025) is given where more comparisons between CARIBIC-AMS and UHSAS are shown.

[Figure]

**Figure 11:** (a) Time series of $O_3$, CO, and particulate sulfate from research flight 4 of the TPEx campaign (June 12, 2024). Shown are the raw data (30 s), a 5-min average, and the detection limit for the 30 s resolution. The lowest panel shows a comparison between the total volume concentration measured by the CARIBIC-AMS and a UHSAS operated in parallel on the Learjet. (b) Tracer-tracer plot with $O_3$ and CO, color-coded by sulfate mass concentration.

Also shown in Fig. 11 (a) is a comparison between the CARIBIC-AMS and data from a UHSAS (Ultra-high sensitivity aerosol spectrometer) which was also operated onboard the Learjet (Joppe et al, 2025) during TPEx. For the comparison, both data sets were averaged over 10 minutes and converted to total volume concentration, assuming spherical particles and an average particle density of 1.5 g cm-3. The agreement is satisfying for altitudes above 8 km, but the noisy signals of organics and ammonium lead to the large scatter of the CARIBIC-AMS data points. As mentioned above, the transmission efficiency of the CPI – ADL combination likely leads to the underestimation by the CARIBIC-AMS at lower altitudes. Further examples of the CARIBIC-AMS – UHSAS comparison during TPEx are given in Joppe et al. (2025).

**Minor comments:**

• Single Ion Calibration: This is a minor point, but I am not following the need for periodic single ion calibrations during flight. The single ion calibration at this point (post-2014) just supplies a scaling factor (the SI) to convert signal to counts per second. Historically, this calibration also included the spectral baseline fit, but that has been automated since 2016 and is taken on a per-run basis. Now, the in-flight SI is needed to scale it to the SI of the sensitivity calibration. BUT once the instrument is running and the SI has been characterized once, the variability of the SI is fully captured by the airbeam variability (especially in an

instrument with a well-working CPI). So while the ability to do this calibration on the fly e.g. in case of in-flight instrument reboot is important and impressive to have implemented, I would really appreciate it if the authors could explain their rationale for why periodic SI calibrations are needed. As a side note, it would be nice to see a figure with either the airbeam or SI change for a couple of flights.

The figure below shows an example of measured airbeam and SI values during a flight sequence (March 2020, the last flight sequence of IAGOS-CARIBIC). It can clearly be seen that the SI values (red points) change over time, and that the airbeam basically follows the SI, but decreases at the beginning of each flight

[Figure]

We assume that this is due to the fact that the initial SI calibration is done right before starting the measurement. At this point the pressure has just dropped below $3\times10^{-6}$ hPa and it is likely that the background signal is still too high and thus the airbeam is high at the beginning and decreases over time during the first phase of each flight.
In the course of all flights, the SI decreases and the airbeam increases again accordingly. We therefore think that an SI calibration later in the flight is more reliable than directly at the beginning, but still the data suggest that it is worthwhile to repeat it several times during a flight.
(For the periodic SI calibration during flight, we use the "SingleIon" routine in the AMS DAQ version 5.0.6)

For a more consistent display of the data sets, we included the AB and SI signal of the October 2019 flight sequence in the new version of Figure 3 (see below).
We added the following sentences to the section "in-flight calibration":

The SI calibration has been conducted during the IAGOS-CARIBIC flights several times during a flight sequence. Figure 03 shows the recorded changes of the SI signal strength during the November 2019 flights, along with the resulting airbeam changes. The airbeam signal refers to the gas-phase signal at m/z 28 ($N_2^+$), which is used for monitoring the sensitivity of the instrument. Note that the airbeam signal decreases after a fresh pump-down, as can be seen in the data of the flight on November 05, 2019. It may therefore be more reasonable to do the SI calibration only later in the flight.

• Figure 3: It would be very helpful if the different operation states described in the text are added to the figure (maybe as shaded bars). Also, please clarify the units for the Yaxis. Are these A at 24 V? It might be clearer to use W(atts) here.

Yes, it's A at 24 V. We changed the units to Watts and added the operation states, the airbeam and the single ion strength to the figure:

[Figure]

• Section 2.2.3 is not a straightforward read, since it tries to combine both topological and process details into one narrative. One suggestion to make it more accessible is to add a simple diagram showing the network/software topology/hierarchy and referring to it in the text.

We tried to summarize section 2.2.3 in a diagram, but could not find a reasonable way to express the processes in a comprehensive way. We therefore left the section basically as it is and changed the text only slightly.

• The units used in the paper are µg m-3 STP. STP is not defined anywhere, and since the atmospheric community is not exactly known for its general adherence to SI units I would strongly urge the authors to spell out what they mean in the text.

STP stands for Standard Temperature (273.15 K or 0°C) and Pressure (1000 hPa). We included a sentence explaining it and the following reference:

'STP' in *IUPAC Compendium of Chemical Terminology*, 5th ed. International Union of Pure and Applied Chemistry; 2025. Online version 5.0.0, 2025.
https://doi.org/10.1351/goldbook.S06036

• Line 19: Instead of "part" maybe "module" would read better?

Changed.

• Line 25: "Due to **the** short time"

Changed.

• Line 86: I would recommend adding the mission description paper for Atom here as well for completeness:
C. R. Thompson, S. C. Wofsy, M. J. Prather, P. A. Newman, T. F. Hanisco, T. B. Ryerson, D. W. Fahey, E. C. Apel, C. A. Brock, W. H. Brune, K. Froyd, J. M. Katich, J. M. Nicely, J. Peischl, E. Ray, P. R. Veres, S. Wang, H. M. Allen, E. Asher, H. Bian, D. Blake, I. Bourgeois, J. Budney, T. Paul Bui, A. Butler, P. Campuzano-Jost, C. Chang, M. Chin, R. Commane, G. Correa, J. D. Crounse, B. Daube, J. E. Dibb, J. P. Digangi, G. S. Diskin, M. Dollner, J. W. Elkins, A. M. Fiore, C. M. Flynn, H. Guo, S. R. Hall, R. A. Hannun, A. Hills, E. J. Hintsa, A. Hodzic, R. S. Hornbrook, L. Greg Huey, J. L. Jimenez, R. F. Keeling, M. J. Kim, A. Kupc, F. Lacey, L. R. Lait, J.-F. Lamarque, J. Liu, K. Mckain, S. Meinardi, D. O. Miller, S. A. Montzka, F. L. Moore, E. J. Morgan, D. M. Murphy, L. T. Murray, B. A. Nault, J. Andrew Neuman, L. Nguyen, Y. Gonzalez, A. Rollins, K. Rosenlof, M. Sargent, G. Schill, J. P. Schwarz, J. M. St. Clair, S. D. Steenrod, B. B. Stephens, S. E. Strahan, S. A. Strode, C. Sweeney, A. B. Thames, K. Ullmann, N. Wagner, R. Weber, B. Weinzierl, P. O. Wennberg, C. J. Williamson, G. M. Wolfe, L. Zeng, THE NASA ATMOSPHERIC TOMOGRAPHY (ATom) MISSION: Imaging the Chemistry of the Global Atmosphere. *Bull. Am. Meteorol. Soc.* **1**, 1–53 (2021).

Added in revised version.

• L161: "**into** one new housing"

Changed.

• L244: "Thus, flow calibrations and size calibration will not change, need to be calibrated only once, and later on only have to be checked.". Agreed. But can you elaborate on how often these checks are done in practice?

This is not defined yet. The goal is to do these checks on a routine basis between the flight sequences. It is currently foreseen that the instrument will be brought back from KIT to MPIC after the IACOS-CARIBIC container is back at KIT after a flight sequence of four or six consecutive flights. The time the instrument remains at our lab is then about 2 or 3 weeks. However, this requires that the personal resources are available, and also assumes that no other instrumental problems have occurred. If no instrumental issues have occurred, the data will be downloaded, the IE calibration, flow and size calibrations checks and other routine checks of instrumental performance will be done.
We added the following sentence to the manuscript:

How frequent these checks have to be performed will depend on the experience from the continued routine operation in future.

• L387: "…than **at** the end of the flight"

Corrected.

**References:**

Bahreini, R., Ervens, B., Middlebrook, A. M., Warneke, C., de Gouw, J. A., DeCarlo, P. F., Jimenez, J. L., Brock, C. A., Neuman, J. A., Ryerson, T. B., Stark, H., Atlas, E., Brioude, J., Fried, A., Holloway, J. S., Peischl, J., Richter, D., Walega, J., Weibring, P., Wollny, A. G., and Fehsenfeld, F. C.: Organic aerosol formation in urban and industrial plumes near Houston and Dallas, Texas, Journal of Geophysical Research: Atmospheres, 114, https://doi.org/10.1029/2008JD011493, 2009.

Brock, C. A., Froyd, K. D., Dollner, M., Williamson, C. J., Schill, G., Murphy, D. M., Wagner, N. J., Kupc, A., Jimenez, J. L., Campuzano-Jost, P., Nault, B. A., Schroder, J. C., Day, D. A., Price, D. J., Weinzierl, B., Schwarz, J. P., Katich, J. M., Wang, S., Zeng, L., Weber, R., Dibb, J., Scheuer, E., Diskin, G. S., DiGangi, J. P., Bui, T., Dean-Day, J. M., Thompson, C. R., Peischl, J., Ryerson, T. B., Bourgeois, I., Daube, B. C., Commane, R., and Wofsy, S. C.: Ambient aerosol properties in the remote atmosphere from global-scale in situ measurements, Atmos. Chem. Phys., 21, 15023-15063, 10.5194/acp-21-15023-2021, 2021.

Drewnick, F., Hings, S. S., Alfarra, M. R., Prevot, A. S. H., and Borrmann, S.: Aerosol quantification with the Aerodyne Aerosol Mass Spectrometer: detection limits and ionizer background effects, Atmos. Meas. Tech., 2, 33-46, 10.5194/amt-2-33-2009, 2009.

Hermann, M., Weigelt, A., Assmann, D., Pfeifer, S., Müller, T., Conrath, T., Voigtländer, J., Heintzenberg, J., Wiedensohler, A., Martinsson, B. G., Deshler, T., Brenninkmeijer, C. A. M., and Zahn, A.: An optical particle size spectrometer for aircraft-borne measurements in IAGOS-CARIBIC, Atmos. Meas. Tech., 9, 2179-2194, 10.5194/amt-9-2179-2016, 2016.

Hu, W., Campuzano-Jost, P., Day, D. A., Croteau, P., Canagaratna, M. R., Jayne, J. T., Worsnop, D. R., and Jimenez, J. L.: Evaluation of the new capture vaporizer for aerosol mass spectrometers (AMS) through field studies of inorganic species, Aerosol Science and Technology, 51, 735-754, 10.1080/02786826.2017.1296104, 2017.

Hu, W., Campuzano-Jost, P., Day, D. A., Nault, B. A., Park, T., Lee, T., Pajunoja, A., Virtanen, A., Croteau, P., Canagaratna, M. R., Jayne, J. T., Worsnop, D. R., and Jimenez, J. L.: Ambient Quantification and Size Distributions for Organic Aerosol in Aerosol Mass Spectrometers with the New Capture Vaporizer, ACS Earth and Space Chemistry, 4, 676-689, 10.1021/acsearthspacechem.9b00310, 2020.

Joppe, P., Schneider, J., Wilsch, J., Bozem, H., Breuninger, A., Curtius, J., Ebert, M., Emig, N., Hoor, P., Ismayil, S., Kandler, K., Kunkel, D., Kurth, I., Lachnitt, H. C., Li, Y., Miltenberger, A., Richter, S., Rolf, C., Schneider, L., Schwenk, C., Spelten, N., Vogel, A. L., Cheng, Y., and Borrmann, S.: Transport of Biomass Burning Aerosol into the Extratropical Tropopause Region over Europe via Warm Conveyor Belt Uplift, EGUsphere, 2025, 1-39, 10.5194/egusphere-2025-1346, 2025.

Knote, C., Brunner, D., Vogel, H., Allan, J., Asmi, A., Äijälä, M., Carbone, S., van der Gon, H. D., Jimenez, J. L., Kiendler-Scharr, A., Mohr, C., Poulain, L., Prévôt, A. S. H., Swietlicki, E., and Vogel, B.: Towards an online-coupled chemistry-climate model: evaluation of trace gases and aerosols in COSMO-ART, Geosci. Model Dev., 4, 1077-1102, 10.5194/gmd-4-1077-2011, 2011.

Liu, P. S. K., Deng, R., Smith, K. A., Williams, L. R., Jayne, J. T., Canagaratna, M. R., Moore, K., Onasch, T. B., Worsnop, D. R., and Deshler, T.: Transmission efficiency of an aerodynamic focusing lens system: Comparison of model calculations and laboratory measurements for the Aerodyne Aerosol Mass Spectrometer, Aerosol Science and Technology, 41, 721-733, 10.1080/02786820701422278, 2007.

Mei, F., Wang, J., Comstock, J. M., Weigel, R., Krämer, M., Mahnke, C., Shilling, J. E., Schneider, J., Schulz, C., Long, C. N., Wendisch, M., Machado, L. A. T., Schmid, B., Krisna, T., Pekour, M., Hubbe, J., Giez, A., Weinzierl, B., Zoeger, M., Pöhlker, M. L., Schlager, H., Cecchini, M. A., Andreae, M. O., Martin, S. T., de Sá, S. S., Fan, J., Tomlinson, J., Springston, S., Pöschl, U., Artaxo, P., Pöhlker, C., Klimach, T., Minikin, A., Afchine, A., and Borrmann, S.: Comparison of aircraft measurements during

GoAmazon2014/5 and ACRIDICON-CHUVA, Atmos. Meas. Tech., 13, 661-684, 10.5194/amt-13-661-2020, 2020.

Middlebrook, A. M., Roya, B., L., J. J., and and Canagaratna, M. R.: Evaluation of Composition-Dependent Collection Efficiencies for the Aerodyne Aerosol Mass Spectrometer using Field Data, Aerosol Science and Technology, 46, 258-271, 10.1080/02786826.2011.620041, 2012.

Molleker, S., Helleis, F., Klimach, T., Appel, O., Clemen, H. C., Dragoneas, A., Gurk, C., Hünig, A., Köllner, F., Rubach, F., Schulz, C., Schneider, J., and Borrmann, S.: Application of an O-ring pinch device as a constant-pressure inlet (CPI) for airborne sampling, Atmos. Meas. Tech., 13, 3651-3660, 10.5194/amt-13-3651-2020, 2020.

Reifenberg, S. F., Martin, A., Kohl, M., Bacer, S., Hamryszczak, Z., Tadic, I., Röder, L., Crowley, D. J., Fischer, H., Kaiser, K., Schneider, J., Dörich, R., Crowley, J. N., Tomsche, L., Marsing, A., Voigt, C., Zahn, A., Pöhlker, C., Holanda, B. A., Krüger, O., Pöschl, U., Pöhlker, M., Jöckel, P., Dorf, M., Schumann, U., Williams, J., Bohn, B., Curtius, J., Harder, H., Schlager, H., Lelieveld, J., and Pozzer, A.: Numerical simulation of the impact of COVID-19 lockdown on tropospheric composition and aerosol radiative forcing in Europe, Atmos. Chem. Phys., 22, 10901-10917, 10.5194/acp-22-10901-2022, 2022.

Schulz, C., Schneider, J., Amorim Holanda, B., Appel, O., Costa, A., de Sá, S. S., Dreiling, V., Fütterer, D., Jurkat-Witschas, T., Klimach, T., Knote, C., Krämer, M., Martin, S. T., Mertes, S., Pöhlker, M. L., Sauer, D., Voigt, C., Walser, A., Weinzierl, B., Ziereis, H., Zöger, M., Andreae, M. O., Artaxo, P., Machado, L. A. T., Pöschl, U., Wendisch, M., and Borrmann, S.: Aircraft-based observations of isoprene-epoxydiol-derived secondary organic aerosol (IEPOX-SOA) in the tropical upper troposphere over the Amazon region, Atmos. Chem. Phys., 18, 14979-15001, 10.5194/acp-18-14979-2018, 2018.

Zahn, A., Obersteiner, F., Gehrlein, T., Neumaier, M., Dyrhoff, C., Förster, E., Sprung, D., Van Velthoven, P., Xueref-Remy, I., Ziereis, H., Bundke, U., Gerbig, C., Scharffe, D., Slemr, F., Weber, S., Hermann, M., Cheng, Y., and Bönisch, H.: IAGOS-CARIBIC MS files collection (v2025.03.25). Zenodo, 2025.

---

## Referee Report (RR1)

My previous review of this manuscript argued that while the technical improvements of the CARIBIC-AMS described therein are very impressive, there was not enough detail provided to actually show that this instrument has promise as a quantitative, autonomous PM sensor in the UT/LS. The revised manuscript has addressed adequately most of these shortcomings but still falls a bit short on the overall quantification front. The authors make a good case that in a lot of ways this is still a case of "COVID-19 hangover", complicated by the general challenges of the CARIBIC platform. Given the overall consistency of the data presented, I am reasonably convinced that the instrument got UT/LS PM right when it was working within a factor of 3 or so if one accounts for the CE, transmission and other uncertainties, which for what the paper aims to show is sufficient. Hence, I recommend publication once a few last comments listed below are addressed. Hopefully after some additional characterizations the next paper will show significant improvements on these fronts, so that the CARIBIC-AMS can truly be a benchmark for CTMs for years to come.

These comments listed below include some responses/clarifications to previously raised points in the discussion which are not germane to the actual review of the revised text, so for simplicity I am going to highlight the actual action items in *italics*:

- *While in the main text it is made clear that the DLs are typical averages under flight conditions, this is not made explicit in the abstract. Also, neither the text nor the abstract make clear that these are CE=1 DLs (I would assume), and that they might change for different ambient conditions. I think both of these points should be added since there is in general a lot of confusion on AMS DLs in general in our community.*
- I would maintain that instrument blanks are the most accurate way to determine/validate detection limits, provided they are taken under realistic conditions. A 2 hour long blank, as described in Drewnick et al (2009), is certainly not realistic, since as shown in that paper the ionizer basically cleans out after 30 min or so, but that does only partially apply to short blanks like the ones presented by the authors. *So I do think that the figure presented in the response showing agreement between the background variability method and the blanks is valuable, and I would ask the authors to consider including it in the paper.*
  Also, to clarify, my previous request for "stratospheric" data was a typo, I just meant high altitude data, I apologize for the sloppy wording.
- *I also think the DL discussion would read better if "noise of the background" would be replaced with "short-term variability of the background". While for high m/z it is indeed electronic/counting noise, this is e.g. certainly not true for ions such as $SO^+$ or $CO_2^+$, which have a high, highly variable, but not necessarily noisy background.*
- *Regarding the presentation of the instrument operation, I want to clarify that what I specifically meant in my previous review was a block diagram showing the control boxes/computers and the connected powerboxes. And then ideally (in a different color/style) which piece of software runs on which computer and what the call hierarchy*

*is. I do think this would help the presentation, which is hard to follow at the moment, especially for e.g. a second year student. Such a diagram would also highlight how certain design choices do improve the overall reliability/fail-safe behaviour of the system, which is a bonus for a paper like this.*

- Regarding ePToF, a couple of clarifications:
    - The time offset for ePToF and PToF is slightly different (see Williams et al, 2016)). Given the different triggering electronics used by the authors it might not be consistent with the Aerodyne findings, but this could be mentioned.
    - While the ePToF dutycycle is indeed higher than PToF, both the spread of the signal over 25-50 bins and the noise in the inversion at low signal levels does result in quite high DLs vs MS mode, so I would not be surprised if regular ePToF operations is ultimately not implemented. Best of luck in any case.
- The inclusion of the inlet loss calculations is very much appreciated, although as the authors state themselves the main uncertainty here is in the exact shape of the AMS lens transmission. Which could be included in Figure 4d for reference (potentially along with some others, e.g. the Molleker 2020 curve in case a different AMS inlet was flown).
- I do understand that in-field calibrations are not an option given the CARIBIC way of doing things. *What I was asking for (possibly in a very confusing way) in my previous comments is for the in-between deployment calibration data. It does sound that instrument optimization efforts resulted in very minimal coverage on that front. Still, if there were e.g. calibrations available every 6 months over the 2018-2020 period, that would I think still be more illuminating than the single point in time comparison provided.*
- Regarding the SI discussion, I think we mostly agree and I have no issues with the current text. What is worth pointing out, however, is that the "rapidly changing airbeam over the first hour of flight" most likely has indeed nothing to do with any inherent sensitivity change (hence the lack of change in the SI) but is just $CO^+$ signal from the high LVOC background in the instrument (if my assessment is correct, the same trend will be observed in the $CO_2^+$ background signal). If this checks out it could be mentioned.
- Regarding the uncertainty of the nitrate comparison in Fig 7a, I don't think I agree with the 35% uncertainty. If this was ambient nitrate (with the usual uncertainties in terms of mixing state, CE and organic/inorganic contributions) that would certainly be the appropriate uncertainty. But here the authors are putting pure AN particles (their analytical standard) into their respective instruments. *So the uncertainty should really come down to the stability of the AMS and the CPC, and not include either RIE or CE, just the overall IE uncertainty. Per Bahreini, that is 10% (explicitly stated in the SI) and should be used here.*
- Regarding the discussion about Collection efficiency (CE), the terms is typically used with two different meanings:

- o Operationally, as in CE = Vchem/Vphys for a given particle sizer. This definition also then includes any transmission/inlet issues and does NOT allow for significant errors on the particle sizer side, since it takes it as ground truth.
  - o Physically, as a defined, rigoruous vaporizer particle bounce correction independent of the agreement with any other instruments. That is what e.g. the Middlebrook et al, 2012 parametrization addresses.

The authors are clearly talking about the operational definition, which for the purposes of proving adequate quantification severely limits which comparisons can be used (only chemical sensors, basically). *Now, based on the UT acidity measurements reported by Nault et al (2021), the UT is almost as acidic (pH<0) as the lower stratosphere. So it would be very surprising if the CE (in the "particle bounce correction" sense) would be anything but 1, not 0.5 for any of the ambient data presented in this paper.* Looking at the other plots in Joppe et al (2025), it does appear that using CE=1 would in general improve the median agreement (as expected) while worsening it in the larger plumes (which is likely related to transmission losses/and or mismatch with the UHSAS size range). Given the challenges, this is a nice finding and supports the overall quality of data of the prototype.

- *One item where a clarification would be appreciated is this sentence in the conclusions: "The time resolution of 30 seconds allows for detection of small-scale spatial and temporal structures on the order of 500 m.". Is this meant to be in the vertical, e.g. while climbing? If so please clarify, because at Mach 0.82 or so (typical cruise speed of a jet airliner), 30 s is about 7 km in the horizontal.*

**References**

A. M. Middlebrook, R. Bahreini, J. L. Jimenez, M. R. Canagaratna, Evaluation of Composition-Dependent Collection Efficiencies for the Aerodyne Aerosol Mass Spectrometer using Field Data. *Aerosol Sci. Technol.* **46**, 258–271 (2012).

R. Bahreini, B. Ervens, A. M. Middlebrook, C. Warneke, J. A. de Gouw, P. F. DeCarlo, J. L. Jimenez, C. A. Brock, J. A. Neuman, T. B. Ryerson, H. Stark, Atlas, E, J. Brioude, A. Fried, J. S. Holloway, J. Peischl, D. Richter, J. Walega, P. Weibring, A. G. Wollny, F. C. Fehsenfeld, Organic aerosol formation in urban and industrial plumes near Houston and Dallas, Texas. *J. Geophys. Res.* **114**, D00F16-D00F16 (2009).

F. Drewnick, S. S. Hings, M. R. Alfarra, A. S. H. Prevot, S. Borrmann, Aerosol quantification with the Aerodyne Aerosol Mass Spectrometer: detection limits and ionizer background effects. *Atmospheric Measurement Techniques* **2**, 33–46 (2009).

S. Molleker, F. Helleis, T. Klimach, O. Appel, H.-C. Clemen, A. Dragoneas, C. Gurk, A. Hünig, F. Köllner, F. Rubach, C. Schulz, J. Schneider, S. Borrmann, Application of an O-ring pinch device as a constant-pressure inlet (CPI) for airborne sampling. *Atmospheric Measurement Techniques* **13**, 3651–3660 (2020).

B. A. Nault, P. Campuzano-Jost, D. A. Day, D. S. Jo, J. C. Schroder, H. M. Allen, R. Bahreini, H. Bian, D. R. Blake, M. Chin, S. L. Clegg, P. R. Colarco, J. D. Crounse, M. J. Cubison, P. F. DeCarlo, J. E. Dibb, G. S. Diskin, A. Hodzic, W. Hu, J. M. Katich, M. J. Kim, J. K. Kodros, A. Kupc, F. D. Lopez-Hilfiker, E. A. Marais, A. M. Middlebrook, J. Andrew Neuman, J. B. Nowak, B. B. Palm, F. Paulot, J. R. Pierce, G. P. Schill, E. Scheuer, J. A. Thornton, K. Tsigaridis, P. O. Wennberg, C. J. Williamson, J. L. Jimenez, Chemical transport models often underestimate inorganic aerosol acidity in remote regions of the atmosphere. *Communications Earth & Environment* **2**, 1–13 (2021).

Williams et al, EPToF Measurements,17[th] AMS Users Meeting, Portland, USA, http://cires1.colorado.edu/jimenez-group/UsrMtgs/UsersMtg17/Williams_UsersMeeting2016_eptof.pdf

---

## Author Response (AR2)

egusphere 2024-3969

Reply to Referee

Reviewer comments in black

Our replies in red

Changes to the text in blue

My previous review of this manuscript argued that while the technical improvements of the CARIBIC-AMS described therein are very impressive, there was not enough detail provided to actually show that this instrument has promise as a quantitative, autonomous PM sensor in the UT/LS. The revised manuscript has addressed adequately most of these shortcomings but still falls a bit short on the overall quantification front. The authors make a good case that in a lot of ways this is still a case of "COVID-19 hangover", complicated by the general challenges of the CARIBIC platform. Given the overall consistency of the data presented, I am reasonably convinced that the instrument got UT/LS PM right when it was working within a factor of 3 or so if one accounts for the CE, transmission and other uncertainties, which for what the paper aims to show is sufficient. Hence, I recommend publication once a few last comments listed below are addressed. Hopefully after some additional characterizations the next paper will show significant improvements on these fronts, so that the CARIBIC-AMS can truly be a benchmark for CTMs for years to come.

> Thank you for this positive rating.

These comments listed below include some responses/clarifications to previously raised points in the discussion which are not germane to the actual review of the revised text, so for simplicity I am going to highlight the actual action items in *italics*:

- *While in the main text it is made clear that the DLs are typical averages under flight conditions, this is not made explicit in the abstract. Also, neither the text nor the abstract make clear that these are CE=1 DLs (I would assume), and that they might change for different ambient conditions. I think both of these points should be added since there is in general a lot of confusion on AMS DLs in general in our community.*

> We now state in the abstract that the stated detection limits are typical averages under flight conditions. However, it has to be emphasized that the values refer to CE = 0.5, because they were inferred from in-flight data. We clarified this in the abstract, in section 3.3 and in Table 1.

- I would maintain that instrument blanks are the most accurate way to determine/validate detection limits, provided they are taken under realistic conditions. A 2 hour long blank, as described in Drewnick et al (2009), is certainly not realistic, since as shown in that paper the ionizer basically cleans out after 30 min or so, but that does only partially apply to short blanks like the ones presented by the authors. *So I do think that the figure presented in the response showing agreement between the background variability method and the blanks is valuable, and I would ask the authors to consider including it in the paper.*

> We included the Figure as Appendix B, and added some explanation in section 3.3:

Detection limits can also be calculated from the automated blank filters as $3 \times \sigma_{Filter}$ ($\sigma$: standard deviation of the calculated mass concentration during the blank measurement; e.g., Drewnick et al., 2009). Figure C1 shows a comparison of both methods for flight 578. The automated blanks were taken every hour for 10 minutes during this flight. In the beginning of the flight, the background signal method yields higher detection limits than the blank filter method, but after about three hours, both methods agree fairly well.

Also, to clarify, my previous request for "stratospheric" data was a typo, I just meant high altitude data, I apologize for the sloppy wording.

- *I also think the DL discussion would read better if "noise of the background" would be replaced with "short-term variability of the background". While for high m/z it is indeed electronic/counting noise, this is e.g. certainly not true for ions such as $SO^+$ or $CO_2^+$, which have a high, highly variable, but not necessarily noisy background.*

We changed to „short-term variability" as suggested

- *Regarding the presentation of the instrument operation, I want to clarify that what I specifically meant in my previous review was a block diagram showing the control boxes/computers and the connected powerboxes. And then ideally (in a different color/style) which piece of software runs on which computer and what the call hierarchy is. I do think this would help the presentation, which is hard to follow at the moment, especially for e.g. a second year student. Such a diagram would also highlight how certain design choices do improve the overall reliability/fail-safe behaviour of the system, which is a bonus for a paper like this.*

We added such a block diagram as Appendix A and refer to it in section 2.2.3.:

A block diagram of the hardware and software components controlling the CARIBIC-AMS operation is shown in Figure A1.

- Regarding ePToF, a couple of clarifications:
o   The time offset for ePToF and PToF is slightly different (see Williams et al, 2016)). Given the different triggering electronics used by the authors it might not be consistent with the Aerodyne findings, but this could be mentioned.
o   While the ePToF dutycycle is indeed higher than PToF, both the spread of the signal over 25-50 bins and the noise in the inversion at low signal levels does result in quite high DLs vs MS mode, so I would not be surprised if regular ePToF operations is ultimately not implemented. Best of luck in any case.

Thank you!

- The inclusion of the inlet loss calculations is very much appreciated, although as the authors state themselves the main uncertainty here is in the exact shape of the AMS lens transmission. Which could be included in Figure 4d for reference (potentially along with some others, e.g. the Molleker 2020 curve in case a different AMS inlet was flown).

We included In Fig. 4d the lens transmission curves from Liu et al (2007) and Knote et a (2011), along with the Molleker et al (2020) curve of the CPI + PM2.5 lens, for 250 hPa (UTLS conditions).

[Figure]

We added the following sentences to the main text:
Also shown in Figure 4 (d) are three transmission functions of different ADL used in the AMS:
The PM1 lens without CPI after Liu et al. (2007) and Knote et al. (2011), and the measurements
at 250 hPa (UTLS conditions) by Molleker et al. (2020) using a CPI and a PM2.5 lens. The overall
transmission from outside to the CPI of the CARIBIC-AMS is above 80% in a size range between
40 and 700 nm. This corresponds well to the size ranges of the AMS inlet systems shown in
Figure 4 (d).

- I do understand that in-field calibrations are not an option given the CARIBIC way of doing
things. *What I was asking for (possibly in a very confusing way) in my previous comments is
for the in-between deployment calibration data. It does sound that instrument optimization
efforts resulted in very minimal coverage on that front. Still, if there were e.g. calibrations
available every 6 months over the 2018-2020 period, that would I think still be more
illuminating than the single point in time comparison provided.*

We added a table (Table B1, Appendix B) with IE calibrations performed during the
IAGOS-CARIBIC operation phase of the CARIBIC-AMS, along with later calibrations
done in the laboratory and after the TPEx campaign in 2024:

| Date | IE (ions molecule$^{-1}$) | AB (s$^{-1}$) | IE/AB (ions (molecule s)$^{-1}$) |
|---|---|---|---|
| June 2019 | 8.44e-8 | 1.88e6 | 5.08e-14 |
| August 2019 | 1.07e-7 | 1.93e6 | 5.38e-14 |
| October 2019 | 1.73e-7 | 2.06e6 | 8.39e-14 |
| November 2019 | 2.01e-7 | 1.94e6 | 1.04e-13 |
| November 2020 | 4.40e-8 | 1.76e6 | 2.50e-14 |
| July/Aug 2024 | 6.23e-8 | 4.09e5 | 1.52e-13 |

- Regarding the SI discussion, I think we mostly agree and I have no issues with the current text. What is worth pointing out, however, is that the "rapidly changing airbeam over the first hour of flight" most likely has indeed nothing to do with any inherent sensitivity change (hence the lack of change in the SI) but is just CO+ signal from the high LVOC background in the instrument (if my assessment is correct, the same trend will be observed in the *CO2+* background signal). If this checks out it could be mentioned.

Yes, your assessment is correct. The $CO_2^+$ background also decreases, thus we agree, the (apparent) airbeam decrease is due to the LVOC background. We clarified that in section 232:

This is due to a background of low volatile organic compounds in the vacuum chamber which slowly evaporate after a fresh pump-down and produce ions such as $CO^+$ (on m/z 28, same as the $N_2^+$ signal) and $CO_2^+$.

- Regarding the uncertainty of the nitrate comparison in Fig 7a, I don't think I agree with the 35% uncertainty. If this was ambient nitrate (with the usual uncertainties in terms of mixing state, CE and organic/inorganic contributions) that would certainly be the appropriate uncertainty. But here the authors are putting pure AN particles (their analytical standard) into their respective instruments. *So the uncertainty should really come down to the stability of the AMS and the CPC, and not include either RIE or CE, just the overall IE uncertainty. Per Bahreini, that is 10% (explicitly stated in the SI) and should be used here.*

We agree and corrected this in Figure 7 and in the text.

[Figure]

- Regarding the discussion about Collection efficiency (CE), the terms is typically used with two different meanings:
o   Operationally, as in CE = Vchem/Vphys for a given particle sizer. This definition also then includes any transmission/inlet issues and does NOT allow for significant errors on the particle sizer side, since it takes it as ground truth.
o   Physically, as a defined, rigoruous vaporizer particle bounce correction independent of the agreement with any other instruments. That is what e.g. the Middlebrook et al, 2012 parametrization addresses.

The authors are clearly talking about the operational definition, which for the purposes of proving adequate quantification severely limits which comparisons can be used (only chemical sensors, basically). *Now, based on the UT acidity measurements reported by Nault et al (2021), the UT is almost as acidic (pH<0) as the lower stratosphere. So it would be very surprising if the CE (in the "particle bounce correction" sense) would be anything but 1, not 0.5 for any of the ambient data presented in this paper.*

Yes, we refer to the operational definition of CE here. We clarified that in the main text in section 3.1:
For the data evaluation, we used an operationally defined constant collection efficiency (CE) of 0.5.
Currently, we have no other available comparison measurement than the aerosol size distribution. We agree that under the high acidity conditions in the UTLS, a CE higher than 0.5 would be expected. For future, more detailed data evaluation we will use the composition dependent CE after Middlebrook et al., 2012, although this may suffer from the high DL for NH4.

Looking at the other plots in Joppe et al (2025), it does appear that using CE=1 would in general improve the median agreement (as expected) while worsening it in the larger plumes (which is likely related to transmission losses/and or mismatch with the UHSAS size range). Given the challenges, this is a nice finding and supports the overall quality of data of the prototype.

*- One item where a clarification would be appreciated is this sentence in the conclusions: "The time resolution of 30 seconds allows for detection of small-scale spatial and temporal structures on the order of 500 m.". Is this meant to be in the vertical, e.g. while climbing? If so please clarify, because at Mach 0.82 or so (typical cruise speed of a jet airliner), 30 s is about 7 km in the horizontal.*

Thank you for spotting this mistake. The 500 m were from another CARIBIC instrument that has 2 sec time resolution. We changed it to 7000 m and rephrased to
„detection of spatial structures on the order of 7000 m".

Additionally, we added a newly available refecence for the TPEx campaign (Bozem et al., 2025) and updated the TPEx data reference (Lachnitt, 2025).

**References**

Bozem, H., Joppe, P., Li, Y., Emig, N., Afchine, A., Breuninger, A., Curtius, J., Hofmann, S., Ismayil, S., Kandler, K., Kunkel, D., Kutschka, A., Lachnitt, H.-C., Petzold, A., Richter, S., Röschenthaler, T., Rolf, C., Schneider, L., Schneider, J., Vogel, A., and Hoor, P.: The TropoPause Composition TOwed Sensor Shuttle (TPC-TOSS): A new airborne dual platform approach for atmospheric composition measurements at the tropopause, EGUsphere [preprint], https://doi.org/10.5194/egusphere-2025-3175, 2025.

A. M. Middlebrook, R. Bahreini, J. L. Jimenez, M. R. Canagaratna, Evaluation of Composition-Dependent Collection Efficiencies for the Aerodyne Aerosol Mass Spectrometer using Field Data. Aerosol Sci. Technol. 46, 258–271 (2012).

R. Bahreini, B. Ervens, A. M. Middlebrook, C. Warneke, J. A. de Gouw, P. F. DeCarlo, J. L. Jimenez, C. A. Brock, J. A. Neuman, T. B. Ryerson, H. Stark, Atlas, E, J. Brioude, A. Fried, J. S. Holloway, J. Peischl, D. Richter, J. Walega, P. Weibring, A. G. Wollny, F. C. Fehsenfeld, Organic aerosol formation in urban and industrial plumes near Houston and Dallas, Texas. J. Geophys. Res. 114, D00F16-D00F16 (2009).

F. Drewnick, S. S. Hings, M. R. Alfarra, A. S. H. Prevot, S. Borrmann, Aerosol quantification with the Aerodyne Aerosol Mass Spectrometer: detection limits and ionizer background effects. Atmospheric Measurement Techniques 2, 33–46 (2009).

S. Molleker, F. Helleis, T. Klimach, O. Appel, H.-C. Clemen, A. Dragoneas, C. Gurk, A. Hünig, F. Köllner, F. Rubach, C. Schulz, J. Schneider, S. Borrmann, Application of an O-ring pinch device as a constant-pressure inlet (CPI) for airborne sampling. Atmospheric Measurement Techniques 13, 3651–3660 (2020).

B. A. Nault, P. Campuzano-Jost, D. A. Day, D. S. Jo, J. C. Schroder, H. M. Allen, R. Bahreini, H. Bian, D. R. Blake, M. Chin, S. L. Clegg, P. R. Colarco, J. D. Crounse, M. J. Cubison, P. F. DeCarlo, J. E. Dibb, G. S. Diskin, A. Hodzic, W. Hu, J. M. Katich, M. J. Kim, J. K. Kodros, A. Kupc, F. D. Lopez-Hilfiker, E. A. Marais, A. M. Middlebrook, J. Andrew Neuman, J. B. Nowak, B. B. Palm, F. Paulot, J. R. Pierce, G. P. Schill, E. Scheuer, J. A. Thornton, K. Tsigaridis, P. O. Wennberg, C. J. Williamson, J. L. Jimenez, Chemical transport models often underestimate inorganic aerosol acidity in remote regions of the atmosphere. Communications Earth & Environment 2, 1–13 (2021).

Williams et al, EPToF Measurements,17th AMS Users Meeting, Portland, USA, http://cires1.colorado.edu/jimenezgroup/UsrMtgs/UsersMtg17/Williams_UsersMeeting2016_eptof.pdf